# Field-scale crop water consumption estimates reveal potential water savings in California agriculture

Anna Boser [1] ✉, Kelly Caylor [1,2], Ashley Larsen [1], Madeleine Pascolini-Campbell [3], John T. Reager [3] & Tamma Carleton [1,4]

Efficiently managing agricultural irrigation is vital for food security today and into the future under climate change. Yet, evaluating agriculture's hydrological impacts and strategies to reduce them remains challenging due to a lack of field-scale data on crop water consumption. Here, we develop a method to fill this gap using remote sensing and machine learning, and leverage it to assess water saving strategies in California's Central Valley. We find that switching to lower water intensity crops can reduce consumption by up to 93%, but this requires adopting uncommon crop types. Northern counties have substantially lower irrigation efficiencies than southern counties, suggesting another potential source of water savings. Other practices that do not alter land cover can save up to 11% of water consumption. These results reveal diverse approaches for achieving sustainable water use, emphasizing the potential of sub-field scale crop water consumption maps to guide water management in California and beyond.

Climate change, drought, and the overexploitation of water resources have led to declines in freshwater storage in many vital agricultural regions[1], raising concerns surrounding the future of food and water security[2,3]. Irrigation constitutes the largest use of freshwater globally[4,5], and many regions will need to decrease their agricultural water use while maintaining high levels of production to ensure a sustainable food and water supply as the population grows[6]. Proposed options for reducing water use include leaving lands fallow, switching to less water-intensive crops, adopting water-saving farming practices such as deficit irrigation, and improving irrigation efficiency[7,8]. However, any approach to reducing agricultural water use depends on the challenging task of characterizing the amount of water crops consume[9].

While it is possible to monitor volumes of water withdrawn for irrigation, this is a poor proxy for the amount of water crops consume through evapotranspiration (ET)[10]. Only some irrigation water results in ET: the rest remains in the system as runoff or recharge, though this proportion can vary widely depending on topography, climate, soil type, and farming practices[11]. Agricultural ET, or the increase in ET that irrigated agriculture brings, is therefore a critical measure in that it represents the amount of water that is actually "consumed" by agriculture. This water leaves the watershed entirely as it is evaporated from the soil and transpired by crops[4]. Since agricultural ET represents agricultural water consumption, it is key for gauging the potential of fallowing, crop switching, or other farming practices to save water[12]. When compared to total irrigation amounts, agricultural ET can also be used to highlight the fraction of irrigation withdrawals that do not effectively result in consumptive use[13]. While this water stays in the system and therefore may continue to provide beneficial uses[14], it also does not achieve its original purpose of contributing to crop growth. Therefore, agricultural ET can help calculate irrigation efficiency and identify unnecessary irrigation water withdrawals from surface or groundwater reservoirs[13].

Two methodological challenges prevent agricultural ET from being monitored at scale. First, simply measuring total ET at scale is a challenge. Second, even when measures of ET are available, it is

[1]Bren School of Environmental Science and Management, UC Santa Barbara, 2400 Bren Hall, Santa Barbara 93106 CA, USA. [2]Department of Geography, UC Santa Barbara, Ellison Hall, Santa Barbara 93106 CA, USA. [3]NASA Jet Propulsion Laboratory, California Institute of Technology, 4800 Oak Grove Drive, Pasadena 91109 CA, USA. [4]National Bureau of Economic Research, 1050 Massachusetts Avenue, Cambridge 02138 MA, USA. ✉e-mail: annaboser@ucsb.edu

difficult to separate agricultural ET from the total[10,15]. To measure total ET, eddy covariance flux towers are highly accurate for monitoring ET at a single location[16]. However, they are expensive and thus sparse, and they are designed to measure ET over uniform vegetation, which is not reflective of complex agricultural landscapes[17]. In the absence of ET measurements, theoretical water demand can be simulated based on climate and crop type[18,19]. Although such tools are helpful for water demand planning[20], theoretical water demand represents the ET of a crop whose water demands are fully met. Therefore, these are likely to overestimate actual ET. Additionally, these simulated estimates exclusively reflect variations in water demand influenced by factors incorporated into the model, which are often limited to crop type and climate[21]. Therefore, analyses based on these models are constrained to examining these specific factors, neglecting other critical drivers of ET such as farming practices. Given the scarcity of in situ data and the inherent constraints of simulated estimates, accurately gauging total ET at scale remains a significant challenge.

Addressing the second challenge, even when accurate ET measurements are available, isolating agricultural ET from total ET is difficult. This step is key for water management, since the resulting metric represents the increase in ET attributable to agriculture. To do so, one must estimate the naturally-occurring ET that would occur in the absence of irrigated agriculture (for example, if the land were left fallow). Most often, this naturally-occurring ET is assumed to be equal to precipitation[19,22]. However, there may be temporal lags between precipitation and resulting ET, some precipitation may not result in ET at all, and ET may also be supplemented by other sources of water, such as near surface groundwater. Together, these challenges leave us with limited measurements of ET and no empirical estimates of agricultural ET at large scales, inhibiting our ability to form evidence-based water management policies that accurately reflect crop water use and potential water savings in agriculture.

Recent advances in the remote sensing of ET unlock new avenues for research into agricultural water consumption. Numerous algorithms for estimating ET using land surface temperature and other remote sensing inputs have been validated specifically for use in agricultural settings[23,24]. These advancements have allowed researchers to empirically study total ET in agricultural settings[25,26]. However, despite high-resolution maps of ET in agricultural landscapes[25,27], the challenge of isolating agricultural ET from total ET has been limited to studies in extremely arid regions where irrigation is the only source of water available to plants[28].

Here, we develop a framework for measuring agricultural ET at sub-field scales. We use remote sensing to determine total ET and combine it with machine learning to estimate naturally-occurring ET. First, we retrieve satellite-based remotely sensed total ET estimates from the 30m, monthly OpenET ensemble data[24] available in the western United States starting in 2016. Second, since we can use these same OpenET estimates to directly observe naturally-occurring ET over fallow lands, we train a gradient boosting algorithm to predict ET over fallow lands. We use information on topography, soil quality, climate, and spatial and temporal coordinates as predictors. We then use the model to retrieve naturally-occurring ET over all active agricultural fields, which we subtract from the remotely-sensed ET to calculate agricultural ET.

We apply this methodology to calculate agricultural ET across California's Central Valley, one of the world's most water-stressed and agriculturally productive regions[29]. We use these maps of agricultural ET to calculate the water consumption of different crop types, as well as to calculate variability in agricultural ET within crop types. These insights allow us to evaluate different strategies that have been proposed to save water under the Sustainable Groundwater Management Act (SGMA)[7], which mandates that all water basins in California reduce groundwater pumping to sustainable levels by 2040[30]. Specifically, we compare the ability of fallowing, crop switching, and other farming practices to save water by reducing agricultural ET. Additionally, we calculate irrigation efficiency to assess the potential to reduce irrigation withdrawals without decreasing agricultural ET. In the wake of groundwater pumping cutbacks of 20–50% under SGMA[7], this work will guide water managers in enacting water savings and help predict which land use changes are likely to ensue.

## Results

### Estimating agricultural ET

We calculate agricultural ET by retrieving the total ET observed over agricultural areas and subtracting naturally-occurring ET (Fig. 1). While total ET estimates are retrieved directly from OpenET, we simulate naturally-occurring ET by training a gradient boosting regressor to predict the ET observed by OpenET over fallow lands. Our model simulating naturally-occurring ET achieves an $R^2$ of .87 and a Mean Absolute Error (MAE) of 35.5 mm per year (Supplementary Fig. 1). We use a test set made up of 2 km² held-out areas (about four times the area of a large agricultural field in the Central Valley). We do not find that error is structured across either space (Supplementary Figs. 2 and 3) or time (Supplementary Fig. 4).

### Variation in agricultural ET within and across crop types

We leverage the significant variability in agricultural ET observed across the Central Valley (Fig. 1) to analyze the factors driving these variations. Crop type, which is commonly cited as an important variable explaining differences in agricultural ET[19], explains 34% of the variation in estimated agricultural ET (Eq. (4), Supplementary Fig. 5). In Fig. 2, we show the water intensity of different crop groups (Supplementary Note 7). Deciduous fruits and nuts are some of the highest consumers at 625 mm per year (582–668 95% CI) (Fig. 2), particularly almonds at 715 mm per year (651–778 95% CI) (Supplementary Fig. 5). Conversely, grain and hay crops consume only 141 mm per year (111–171 95% CI). These broad variations in the water consumption of different crop types align with previous work estimating crop water demands (Supplementary Note 1, Supplementary Fig. 6) and suggest that substantial water savings may be possible with crop switching.

However, the majority of the variation remains unexplained by crop type (Fig. 2, Supplementary Fig. 5). This within-crop variability is not uniform for different crops: crops such as deciduous fruits and nuts tend to have large variances, whereas rice has a very small variance. For example, pistachios have the most variability with an interquartile range (IQR) of 664 mm per year over the whole state. Conversely, rice has an IQR of only 122 mm per year (Supplementary Fig. 5).

Part of the large variability in agricultural ET for deciduous fruits and nuts can be explained by orchard age: young orchards consume significantly less water than more mature ones (Supplementary Note 2, Supplementary Fig. 7). Climate, topography, and soil quality explain an additional 6% of within-crop variation. However, substantial within-crop variation remains, indicating that some of these differences may be due to variations in farming practices. This would suggest that reducing water consumption without switching crops may be feasible, which we explore in the next section.

### The water-saving potential of different management strategies

The variability of agricultural ET both within and between crops allows us to evaluate the water saving-potential of different management strategies. Here, we compare the effect of three scenarios on reducing agricultural ET in groundwater sub-basins across the Central Valley:

1. Crop switching: Substitute high-ET crops for the median water-consuming crop (Eq. (6)).
2. Farming practices: Keep the same spatial allocation of crops, but reduce agricultural ET of high consumers to the median, crop-specific, consumption level (Eq. (7)).
3. Fallowing: Fallow the 5% of lands with the highest estimated agricultural ET (Eq. (8)).

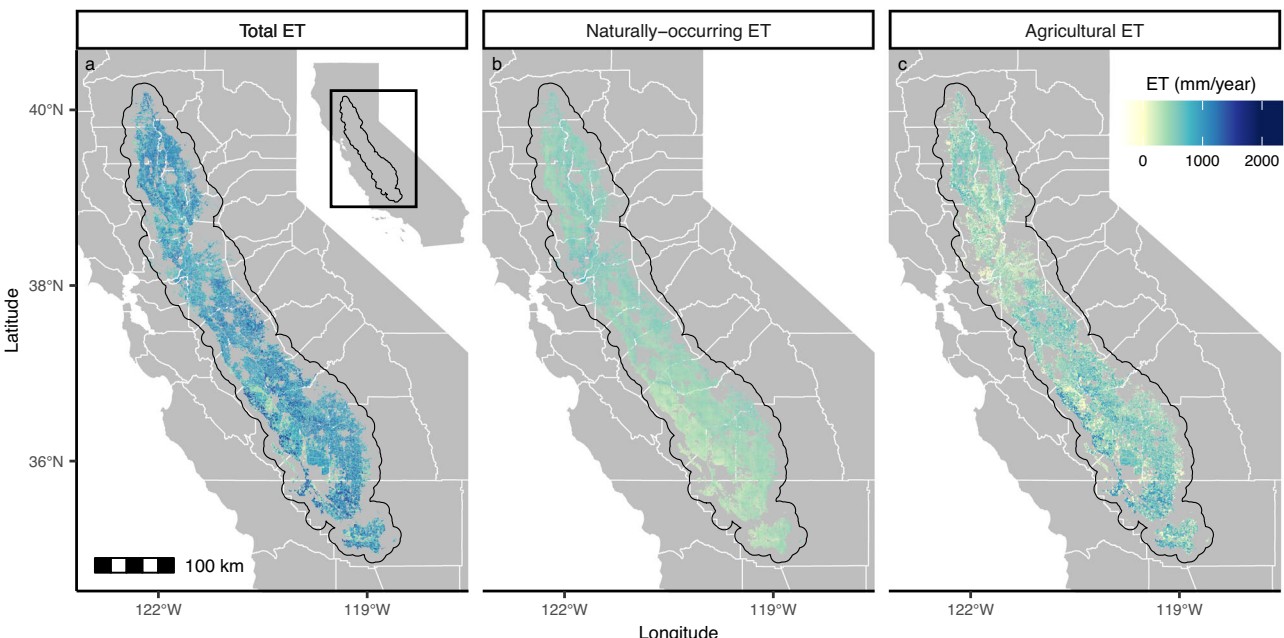

**Fig. 1 | Estimates of annual agricultural evapotranspiration (ET) over active agricultural lands in California's Central Valley. a** Total ET is remotely sensed and is retrieved from OpenET. **b** Naturally-occurring ET is an estimate of the ET that would be present if the agricultural lands were fallow, and is predicted using machine learning. **c** Agricultural ET is the difference between total and naturally-occurring ET, and represents our estimate of the ET caused by agriculture, and therefore the water that would be conserved if the land were fallow instead of cropped. The variations in agricultural ET across the landscape suggest that different fields can have vastly different abilities to conserve water. OpenET provides ET data at the scale of 30 m; all figure panels here show ET resampled to 70 m resolution for computational efficiency and to better match average field size.

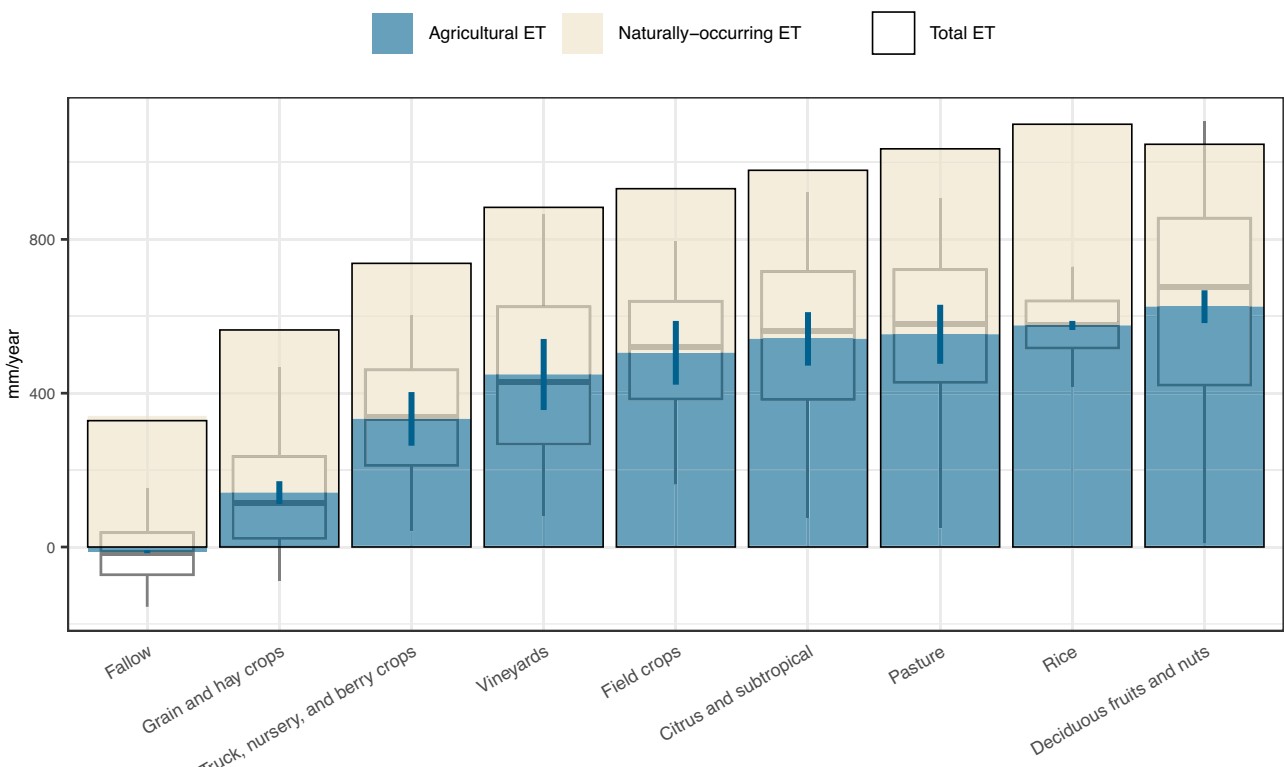

**Fig. 2 | Variations in annual agricultural evapotranspiration (ET) across and within crop groups.** Mean agricultural ET by crop group (blue fill and 95% CI) is the average difference between total ET (black outline) and naturally-occurring ET (cream fill). All measures are summed across the year, leading to naturally-occurring and total ET estimates that include water consumption occurring outside of the growing season. While we find significant differences in mean agricultural ET across crop groups, the gray box plots also show a broad spread in agricultural ET within crop groups (box plots show 0.5, 0.25. 0.5, 0.75, and 0.95 quantiles).

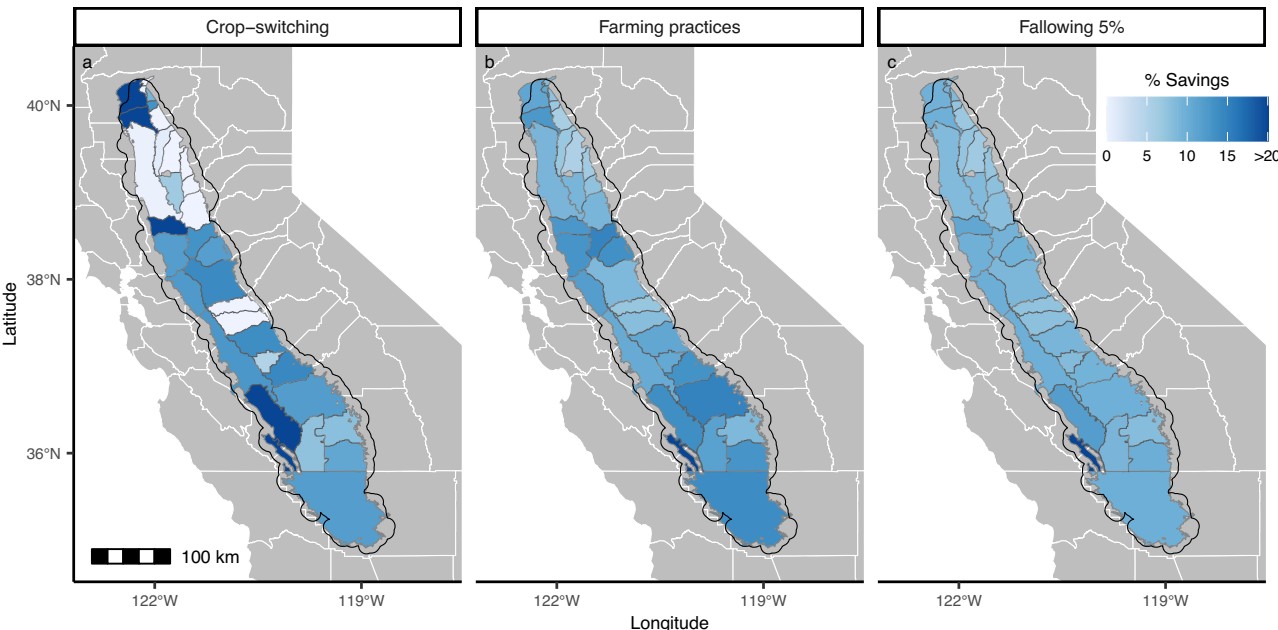

**Fig. 3 | The percent reduction in agricultural evapotranspiration (ET) driven by various management scenarios. a** Savings accrued by substituting high-ET crops for the median water-consuming crop in a sub-basin. **b** Water savings without changing land cover by reducing agricultural ET of high consumers to the median, crop- and sub-basin-specific, consumption level. **c** Water savings from fallowing the 5% of lands with the highest estimated agricultural ET.

Because it would not be realistic to prescribe farmers in different contexts to consume similar amounts of water, we calculate crop-specific agricultural ET and within-crop variation at the level of groundwater sub-basins. We additionally control for differences in climate, topography, and soil type before conducting the scenarios. To account for the effects of orchard age, we remove orchards that have been bearing fruit for 5 years or less or that are in their last year of production (Supplementary Fig. 7).

We find that each of these scenarios return similar reductions in overall water consumption of around 10% (Fig. 3). Since crops with high agricultural ET are heavily favored in most sub-basins and therefore the median crop usually has a high agricultural ET, crop switching only results in a 9.9% reduction in agricultural ET. In order to achieve greater water savings, crops would need to be switched to less popular crops with lesser water demands: switching all crops to the lowest consuming crop in a sub-basin results in a 93.8% reduction in agricultural ET. While strategies that do not require a change in land cover type are less commonly cited as a water management approach, we find that reducing high consumers to their crops's median consumption level yields similar water savings of up to 11.3%. Fallowing does, however, remain the most effective way to reduce agricultural ET. We find that leaving a mere 5% of land fallow results in a 9.3% consumption reduction, comparable to the savings afforded by crop switching and within-crop scenarios which by definition affect up to 50% of lands.

### Irrigation efficiency
While fallowing, crop switching, and other farming practices highlight opportunities to decrease agricultural ET, we also find opportunities to diminish runoff and deep percolation during conveyance, on-farm management, or application (Fig. 4). On average, we find that irrigation in the Central Valley is 61.8% (54.0–69.7%) efficient, similar in magnitude to what theoretical estimates predict (Supplementary Note 3, Supplementary Fig. 8). The large disparity in efficiency between northern and southern counties is, however, more pronounced than theoretical estimates (Supplementary Fig. 8). As a result, there may be limited potential to decrease water withdrawn for irrigation in the

south which achieves efficiencies as high as 80%. The particularly low efficiencies in the northern counties, however, suggest a large potential to decrease irrigation water withdrawals without affecting agricultural ET.

## Discussion
By empirically quantifying crop water consumption at sub-field scales, we contribute to characterizing agriculture's hydrological effects and evaluate the ability of different management strategies to mitigate this impact. To overcome the methodological challenges associated with estimating agricultural ET, we leverage recent advances in remotely sensed ET and use machine learning to generate a naturally-occurring ET counterfactual. The fine scale variability in agricultural ET we uncover allows us to analyze the drivers of these differences and simulate the potential for different management practices to save water, including under-explored ones like adjusting farming practices. Additionally, when comparing agricultural ET to total water withdrawn for irrigation, we find substantial opportunities for improvements in irrigation efficiency in the northern counties of the Central Valley. High-resolution maps of agricultural ET can therefore guide our understanding of how agriculture and management can affect water resources in California and other water stressed agricultural regions globally.

Accurate agricultural ET measures are crucial to characterize anthropogenic impacts on the hydrological cycle[31] and enact effective water management[18,32]. Previously, agricultural ET has been estimated by (i) simulating crop water demand based on crop type and climate and (ii) removing naturally available water by subtracting precipitation[19,22]. However, both of these steps embed assumptions that can lead to significant biases. For (i), simulated crop water demand may not adequately represent the field characteristics or farming practices of a given location[13]. For example, water demands are not necessarily always met, meaning water consumption may be overestimated[33–35]. In line with this, while we generally find good agreement with crop water demand simulated by the CalSIMETAW model[21], we do find that agricultural ET estimates are significantly smaller (Supplementary Note 1, Supplementary Fig. 6). For (ii), using

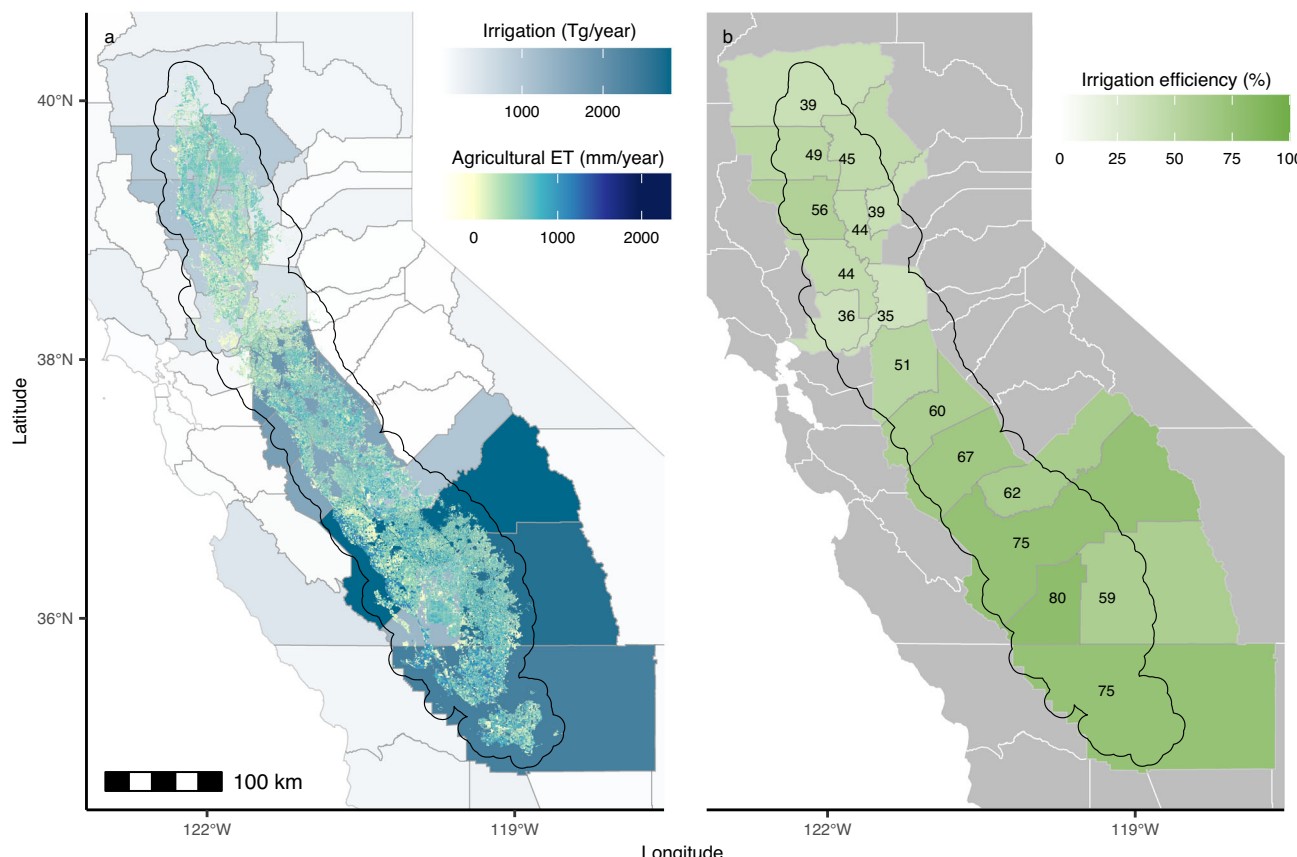

**Fig. 4 | Irrigation efficiency across the counties of the Central Valley. a** Irrigation efficiency is calculated by dividing agricultural evapotranspiration (ET) (gridded data) by USGS county-level reports of irrigation amounts (blue polygons). For the calculation, agricultural ET is averaged to the county level to match the spatial scale of the irrigation data. Additionally, irrigation is displayed in volumetric units (teragrams), but is divided by county-level cropland area to be in units consistent with agricultural ET prior to the calculation. We note that these irrigation amounts are counted at the point of use, rather than the water's point of origin. **b** The resulting county-level irrigation efficiency estimates vary widely across the Central Valley, with particularly low efficiencies in the northern counties.

precipitation to proxy for naturally-occurring ET fails to account for precipitation being lost as runoff, temporal lags in when precipitation is transpired, and alternative natural sources of water such as shallow groundwater. We find that the bias between our estimates and CalSI-METAW estimates increases when precipitation, rather than naturally-occurring ET, is used to represent baseline amounts of available water. This can be explained by the seasonal mismatch between precipitation and naturally-occurring ET: although annually there is more pre-cipitation than naturally-occurring ET, precipitation mainly occurs in the winter when it is unavailable to most crops. Because using pre-cipitation as a proxy for naturally-occurring ET does not account for moisture that remains in the soil by the time the growing season begins, this inflates simulated estimates of crop water demand. Our use of remotely sensed ET estimates and our ability to estimate naturally-occurring ET therefore allows us to more accurately char-acterize agricultural ET.

High-resolution, empirical estimates of agricultural ET addition-ally open up the possibility to investigate differences in agricultural ET beyond what is attributable to crop type, which we find only accounts for 34% of the variation. These within-crop variations can be sub-stantial for many crops: the difference in agricultural ET of a pistachio field from the 75th to the 25th percentile is the same as the water that could be saved from fallowing an alfalfa field. Such broad variability is consistent with findings from studies comparing total ET across crops during the growing season[25,26]. Even after adjusting for variability attributable to field characteristics, orchard age, and climate, we find that these variations could translate to substantial water savings

without requiring a change in crop type. Unfortunately, a lack of high-resolution data on field-scale farming practices and yields inhibits further analysis of specific practices driving agricultural ET variation and their economic implications. However, prior literature from experimental plots or particular locations suggests that mulching[36], conservation tillage[37], deficit irrigation[33–35], and improved irrigation scheduling and technologies[38] all have potential to limit agricultural ET. These practices may prove advantageous relative to costly strate-gies like crop-switching or fallowing, though more detailed cost-benefit analyses are necessary to determine the suitability of various interventions in specific contexts[39]. Such water saving farming prac-tices are not mentioned in the plans drafted by SGMA water managers[7]. This is possibly due to a lack of conclusive research on the potential of such strategies to effectively decrease agricultural ET without sig-nificant effects on yield or operation cost. Spatial data on the use of these different practices would allow researchers to take full advan-tage of our high-resolution agricultural ET estimates and study their water-saving and economic benefits.

In situ assessments of crop-specific water consumption and other variations in agricultural ET enable us to model potential water savings across diverse management scenarios. In addition to previously men-tioned scenarios based on farming practices, we also investigate more traditionally studied strategies, such as crop switching and fallowing. Consistent with prior studies relying on crop water demand simula-tions, our research suggests that transitioning to crops with lower water requirements is an effective conservation strategy. However, accruing substantial water savings requires embracing less popular

crops like grains and hay[20]. The feasibility of increasing production of these crops in the Central Valley is uncertain due to high labor and operational costs[20]. Furthermore, any form of crop switching entails expenses related to the adoption of new knowledge, technologies, and market adjustments[40,41]. Consequently, the viability of a significant shift towards less water-demanding crops, and its alignment with market expansion, remains uncertain[20]. Therefore, our findings support the notion that extensive fallowing or land retirement may be essential to achieve substantial reduction targets in areas with severe overdrafts[42,43]. Considering the risk of increased dust from unused land, repurposing such areas for habitat restoration, flood water capture for groundwater replenishment[44], solar energy production, or sustainable industrial development[45] could mitigate the economic impacts of land retirement for both farmers and local communities. Detailed agricultural ET maps like the ones generated for this study can help determine the scale of land repurposing needed and identify priority areas for such initiatives under different constraints, including existing water rights[46].

Finally, we find a significant opportunity to improve irrigation efficiency, especially in the northern part of the Central Valley where we find lower irrigation efficiency than previously expected (Supplementary Note 2, Supplementary Fig. 7). This implies that farmers in these areas could potentially reduce their irrigation water usage without negatively impacting agricultural ET and, consequently, crop growth. Although this finding is consistent with management strategies that focus on boosting irrigation efficiency[7], it is important to recognize that such improvements may not automatically lead to water savings at the watershed level[14]. This is because water not consumed in irrigation processes does not always exit the watershed but can instead be reabsorbed into groundwater reservoirs or surface water bodies for later reuse. Conversely, water that is evapotranspired, which constitutes the 'efficient' part of irrigation, is completely removed from the watershed[14]. Hence, paradoxically, increasing irrigation efficiency could reduce water availability if it is not matched by a decrease in the amount of water withdrawn for irrigation. To prevent an unintended increase in water consumption, managers could use agricultural ET maps to monitor and control water use as improvements in irrigation efficiency are implemented.

This study has some important limitations. Most notably, due to a lack of data on farming practices, it is difficult to ascertain whether the variation in within-crop agricultural ET that we estimate is indeed due to farming practices. When conducting our management scenarios, we account for the effects of climate, soil quality, topography, and orchard age. We additionally conduct the scenarios at the level of small groundwater sub-basins rather than across the entire valley to account for any additional regional environmental differences we are unable to otherwise account for. However, variance stemming from error in our agricultural ET estimates or from mislabeled crop types could contribute to observed within-crop variation in agricultural ET. We minimize error from mislabeled crop types by using the most accurate crop data available in California which boasts an accuracy of 97.6%[47]. Additionally, the OpenET ensemble model has been extensively validated[24,48], and our machine learning model has an $R^2$ of .87. We estimate that these sources of error are responsible for only 11% of the variance in our yearly agricultural ET estimates (Supplementary Note 4). Nevertheless, the water-saving potentials we calculate for both the fallowing and farmers practice scenarios should be interpreted as upper bound estimates.

Another important limitation of this study stems from the limited data available on water withdrawn for irrigation which is needed to calculate irrigation efficiency. Since irrigation data are not available over the same years as agricultural ET, we use the average of the two most recent years, 2010 and 2015, a drought year and non-drought year, to calculate irrigation efficiency. To ensure that our results are robust to year-to-year variations in agricultural ET and irrigation use,

we calculate irrigation efficiency using all the possible combinations of years (Supplementary Fig. 9). Though we do find significant variations based on the irrigation year used in some counties, we consistently find low irrigation efficiencies in the north and higher ones in the south. Irrigation data with better spatial and temporal resolution would improve estimates of irrigation efficiency calculated using this method.

We have shown how highly resolved agricultural ET estimates can improve our understanding of anthropogenic impacts on the hydrologic cycle and guide water management by quantifying the potential of different water-saving strategies. In California, our findings suggest that irrigated agriculture increases ET less than previously estimated, and we find that managers may not need to rely as heavily on changes in land cover as currently proposed to achieve significant water savings. Because our method for estimating agricultural ET is based entirely on remote sensing and machine learning, it remains cost effective and has the potential to be used globally, especially as global ET datasets become increasingly available[49]. This work can help refine our understanding of agriculture's effect on water resources and help managers achieve water-saving goals in water-stressed agricultural landscapes across the globe.

## Methods
### Agricultural ET
We define agricultural ET as the difference between total ET over an agricultural parcel and the ET that would have been, had that parcel been fallow land instead (Eq. (1)). This definition recognizes that not all ET over agricultural lands, during the growing season or otherwise, can necessarily be attributed to agriculture. Such a definition is particularly useful from a management perspective, since it denotes the decrease in ET, or water savings, that one might expect if agriculture were to cease.

$$ET_{ag} = ET_{tot} - ET_{nat} \qquad (1)$$

where $ET_{ag}$ is agricultural ET, $ET_{tot}$ is the total ET over an agricultural parcel, and $ET_{nat}$ is the counterfactual ET that would occur naturally, were the same land fallow.

We note that another, theoretically distinct counterfactual could be constructed to represent $ET_{nat}$: ET if the land were undisturbed, natural land rather than fallow. We elect to simulate ET over fallow lands since it allows us to predict the potential water savings from fallowing, which has a management relevant interpretation.

**Total ET estimates.** We construct $ET_{tot}$ from OpenET ensemble data, available at a monthly time step from 2016-2021. This data is calculated as the average output of six different ET models estimated using Landsat data[24]. The data have been corrected for biases in ET resulting from data only being available during cloudless overpasses and have been extensively evaluated over a broad variety of land covers[48]. While they are published at a 30m scale, we resample to 70m to improve computational efficiency since this is well below the average size of an agricultural field in the Central Valley.

**Calculation of a naturally-occurring ET counterfactual using machine learning.** We estimate naturally-occurring ET for all agricultural pixels using a gradient boosting regressor trained on data from fallow lands. We choose a gradient boosting regressor due to its high flexibility in learning non-linear relationships and proven performance on tabular datasets relative to other regressors[50]. We retrieve the locations of fallow fields in the Central Valley from the Department of Water Resources (DWR) statewide crop mapping dataset over available years: 2016, 2018, and 2019. Pixels that are within the top 5% ET during July-September are removed due to some implausibly high numbers during these months that suggest mislabeling of an active

agricultural pixel as fallow. While we believe that this data cleaning step is important to ensure an unbiased training set, omitting this step does not significantly change our main findings (Supplementary Note 5, Supplementary Figs. 10–12). We additionally find that our findings do not change when training our model on data that are marked fallow by both the DWR dataset and the Cropland Data Layer (CDL) (Supplementary Note 6, Supplementary Figs. 13–15).

We predict naturally-occurring ET based on latitude and longitude, the month and year, as well as a broad set of additional variables describing topography (elevation, aspect, slope, topographic wetness index), soil quality (California Storie Index), and climate (Potential ET). The latitude, longitude, and month and year indicators are included to capture the spatial and temporal patterns in ET underlying the densely distributed fallow fields in our dataset. However, the inclusion of an indicator variable for each year in our model limits our analysis to 2016, 2018, and 2019, as these are the only years for which we have available land cover data. Nevertheless, given that 'year' contributes a mere 3% to our model's predictive capability (see Supplementary Fig. 17), it may be considered non-essential for future research in this area.

Including additional predictors in our model presents two benefits. First, they can improve the model's predictive power. For example, we find that Potential ET contributes greatly to the model's final predictions (Supplementary Fig. 17). Other variables are important to include because they can help correct for systematic differences between the fallow lands used to train our model and the active agricultural lands we apply the model to. Such differences could arise from farmers selecting lands to be fallowed due to their inherently lower productivity, which would negatively bias our estimates of counterfactual naturally-occurring ET in locations actively being cropped today. To assemble these predictor variables, we retrieve topographic information from the USGS National Elevations Database, soil quality information from the California Storie Index in the the USDA's gSSURGO and STATSGO2 datasets, and Potential ET from the hPET global dataset[51].

To validate our naturally-occurring ET model, we split our dataset, reserving 60% for training, 10% for validation, and 30% for testing. In order to ensure that nearby and therefore very similar pixels are not present across multiple splits, we group our splits by 2 km² squares, four times the size of a large agricultural field in the Central Valley. The entire dataset is made up of over 16 million pixels populating 8180 distinct 2 km² regions. We set aside 4908 of these 2 km² clusters for training, amounting to nearly 10 million pixels. The testing split is composed of 2454 clusters. We tune hyperparameters using 100 iterations of a threefold randomized search cross-validation on an unclustered subset of our dataset. Randomized search cross-validation is similar to grid search cross-validation, but only reaches a random subset of the possible hyperparameter combinations to improve computational efficiency. We manually set the minimum number of samples required to split a node to 200 and the minimum number of samples required at each leaf node to 100 to account for the large size and spatial clustering of the dataset. Because we carry out our analyses using yearly estimates of agricultural ET, we validate our model on yearly estimates of naturally-occurring ET. Only fallow lands from the test data split are used in any subsequent analyses on fallow lands.

## Analysis

After computing agricultural ET for all fields across the Central Valley following Eq. (1), we report a variety of statistical summaries and conduct scenarios manipulating the spatial distribution of agricultural ET.

In order to ensure our analyses only capture variations across space, we aggregate our observations of agricultural ET to a yearly timestep and control for variation across years before beginning this analysis. To do so, we calculate an "adjusted" agricultural ET for each pixel by removing the difference between the mean agricultural ET for that year and the overall sample mean agricultural ET from the original agricultural ET calculation, as follows:

$$\text{AdjustedET}_{p,y} = \text{ET}_{p,y} - (\overline{\text{ET}_y} - \overline{\text{ET}}) \tag{2}$$

where $\text{AdjustedET}_{p,y}$ is the agricultural ET for pixel $p$ in year $y$ with the year-specific variation removed, $\text{ET}_{p,y}$ is the original agricultural ET, $\overline{\text{ET}_y}$ is the mean agricultural ET in year $y$ across all pixels and $\overline{\text{ET}}$ is the mean agricultural ET across all pixels and years.

**Regressions.** To calculate point estimates and confidence intervals for the overall mean agricultural ET (Eq. (3)), the mean agricultural ET by crop group or crop type (Supplementary Note 7), or the mean agricultural ET by county (used to calculate irrigation efficiency) (Eq. (4)), we conduct a series of regressions with the following format:

$$\text{AdjustedET}_{p,y} = \alpha + \epsilon_{p,y} \tag{3}$$

$$\text{AdjustedET}_{p,y} = \beta_{p,y} \times \text{Group}_{p,y} + \epsilon_{p,y} \tag{4}$$

where $\text{AdjustedET}_{p,y}$ represents agricultural ET for pixel $p$ in year $y$ with the year-specific variation removed, $\alpha$ is a coefficient indicating the mean agricultural ET across all samples, $\text{Group}_{p,y}$ represents a vector of dummy variables indicating which crop (for crop comparisons shown in Figs. 2 and S5) or county (for irrigation efficiency calculations shown in Fig. 4; county does not vary by year) observation $p, y$ falls into, $\beta_{p,y}$ is a corresponding vector of coefficients indicating the average agricultural ETs for each crop or county group, and $\epsilon_{p,y}$ is an error term. Standard errors are calculated using 75 km clusters, as determined by plotting a variogram of the spatial autocorrelation in agricultural ET (Supplementary Fig. 16).

The aggregate crop group (Fig. 2) and detailed crop type (Supplementary Fig. 5) are each assigned using the same DWR land cover data used to determine fallow status.

In addition to using regression to calculate point estimates and confidence intervals, regression allows us to calculate the proportion of variation that is explained by a set of variables. This is because the $R^2$ corresponds to the fraction of variation explained by the regression. We use this to assess the % variation explained by crop type, using Eq. (4), as well as that explained by climate, topography, and soil quality:

$$\text{AdjustedET}_{p,y} = \alpha + \beta_{p,y} \times X_{p,y} + \epsilon_{p,y} \tag{5}$$

where $\alpha$ is the intercept, $X_{p,y}$ is a vector of variables including PET, elevation, aspect, slope, TWI, and soil quality, $\beta_{p,y}$ is a vector of coefficients for each variable in $X_{p,y}$, and $\epsilon_{p,y}$ is an error term.

**Management scenarios.** We conduct three scenarios in which we manipulate agricultural ET to mimic land management changes and report the decrease in overall agricultural ET within groundwater sub-basins across the Central Valley. Eq. (6) is a crop switching scenario (*cs*) where pixels with agricultural ET values above those of the median crop in a given sub-basin are replaced with that median crop. Eq. (7) is a farming practice scenario (*fp*) where pixels with agricultural ET values above crop and sub-basin-specific mean values are replaced with that crop and sub-basin mean. Finally, Eq. (8) is a fallowing scenario (*fal*) which replaces all pixels with agricultural ET values above the sub-basin-specific 95th percentile with a value of zero agricultural ET.

$$\text{Savings}_{cs} = 1 - \frac{\sum_{i=1}^{n} \min(\text{ET}_{c,b}, \text{Median}(\text{ET}_{c,b})_b))}{\sum_{i=1}^{n} \text{ET}_i} \tag{6}$$

$$\text{Savings}_{\text{fp}} = 1 - \frac{\sum_{i=1}^{n} \min(\text{ET}_i, \text{ET}_{c,b})}{\sum_{i=1}^{n} \text{ET}_i} \qquad (7)$$

$$\text{Savings}_{\text{fal}} = 1 - \frac{\sum_{i=1}^{n} \begin{cases} \text{ET}_i, & \text{if } \text{ET}_i \leq \text{ET}_{.95,b} \\ 0, & \text{otherwise} \end{cases}}{\sum_{i=1}^{n} \text{ET}_i} \qquad (8)$$

where $\text{ET}_i$ is the adjusted agricultural ET for observation $i$, $\text{ET}_{c,b}$ is the average agricultural ET for crop type $c$ in groundwater sub-basin $b$, $\text{Median}(\text{ET}_{c,b})_b$ is the median $\text{ET}_{c,b}$ in sub-basin $b$, and $\text{ET}_{.95,b}$ is the 95th percentile of agricultural ET in sub-basin $b$.

To ensure that the water savings identified in our scenarios result from factors that can be influenced by management interventions, we account for orchard age, climate, and other physical characteristics of the land. To account for orchard age, we remove young orchards that are have been bearing fruit for 5 or less years, or old orchards that are going to be removed within the next year (Supplementary Note 2). Because we are only able to label orchard age in this way for all orchards in year 2019 (the DWR crop type data we use are only available starting 2014), we exclusively use 2019 for this part of the analysis. Since we are only using one year, we do not adjust the water use according to the year like we do for the other analyses. To account for differences in climate and other characteristics inherent to the land, we control for potential ET, soil quality, topographic wetness index, elevation, aspect and slope using linear regression (Eq. (5)).

**Irrigation efficiency.** We define irrigation efficiency as the proportion of irrigation water that results in agricultural ET, and is thus consumed by agriculture (Eq. (9)). To determine amounts of water withdrawn for irrigation, we retrieve county level irrigation water use data from the USGS National Water Information System. We note that these irrigation amounts are counted at the point of use, rather than the water's point of origin. These data are gathered every five years with the most recently available data from 2015. Because there is no match for the years of available irrigation data and our agricultural ET estimates from 2016, 2018, and 2019, we use the two most recent years, 2010 and 2015, which represent a non-drought year and a drought year, respectively.

$$\text{IRRIGATION EFFICIENCY} = \frac{\text{ET}_{\text{ag}}}{\text{Irrigation}} \qquad (9)$$

where $\text{ET}_{\text{ag}}$ is agricultural ET (Eq. (1)) and Irrigation is the water withdrawn for irrigation. $\text{ET}_{\text{ag}}$ and Irrigation must be in matching units, either volumetric or depth. We calculate both in mm per year.

We assume all active agricultural lands in the Central Valley are irrigated and calculate the average agricultural ET in mm per year over active agricultural lands in each county. To also retrieve average irrigation amount across irrigated lands in a county in mm per year, we divide the volume of irrigation water by the average area of irrigated land in each county. Because some counties are not fully encompassed within the Central Valley, we assume that irrigation in a given county is evenly distributed over irrigated lands within and outside of the Central Valley.

## Data availability
The annual, 70m total, agricultural and naturally-occurring ET data generated in this study have been deposited in the Annual, field-scale crop water consumption estimates database under accession code https://doi.org/10.6084/m9.figshare.24600240. The OpenET data used in this study are available in the Google Earth Engine database under accession code https://developers.google.com/earth-engine/datasets/catalog/OpenET_ENSEMBLE_CONUS_GRIDMET_MONTHLY_v2_0. The crop type data used in this study are available in the Statewide Crop Mapping database under accession code https://data.

cnra.ca.gov/dataset/statewide-crop-mapping. The Cropland Data Layer (CDL) crop type data used in this study are available in the United States Department of Agriculture National Agricultural Statistics Service database under accession code https://www.nass.usda.gov/Research_and_Science/Cropland/Release/index.php. The potential evapotranspiration data used in this study are available in the Hourly potential evapotranspiration (hPET) at 0.1degs grid resolution for the global land surface from 1981-present database under accession code https://data.bris.ac.uk/data/dataset/qb8ujazzda0s2aykkv0oq0ctp. The topography data used in this study are available in the Elevation in the Western United States (90 meter DEM) dataset under accession code https://www.sciencebase.gov/catalog/item/542aebf9e4b057766eed286a. The county shapefile data used in this study are available in the US Census TIGER dataset under accession code https://www2.census.gov/geo/tiger/GENZ2018/shp/cb_2018_us_county_500k.zip. The irrigation data used in this study are available in the USGS Water Data for California database under accession code https://waterdata.usgs.gov/ca/nwis/. The CalSIMETAW data used in this study are available in the Cal-SIMETAW Unit Values database under accession code https://data.ca.gov/dataset/cal-simetaw-unit-values.

## Code availability
All code necessary to replicate this study can be found at the https://doi.org/10.5281/zenodo.10578652 and GitHub repository https://github.com/anna-boser/ET_ag_OpenET.

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

## Acknowledgements

We thank Kathy Baylis for her help conceptualizing the methods. A.B. was supported by the National Science Foundation Graduate Research Fellowship Program (1650114) and the Kuni Foundation. K.C. was supported by the National Science Foundation under Grant DEB-1924309 and ITE-2236021 and the Zegar Family Foundation. A.L. was supported by the National Science Foundation under Grant DEB-2042526. Use was made of computational facilities purchased with funds from the National Science Foundation (CNS-1725797) and administered by the Center for Scientific Computing (CSC). The CSC is supported by the California NanoSystems Institute and the Materials Research Science and Engineering Center (MRSEC; NSF DMR 1720256) at UC Santa Barbara. A portion of this work was conducted at the Jet Propulsion Laboratory, California Institute of Technology, under contract with NASA.

## Author contributions

A.B. collected the data, performed the analyses, and wrote the original draft. A.B., T.C., and K.C. developed the methodology used. A.B., T.C.,

A.L., and K.C. reviewed and edited the draft. A.B., T.C., A.L., K.C., J.R., and M.P. contributed to the conceptualization of the study.

## Competing interests

The authors declare no competing interests.

## Additional information

**Peer review information** : *Nature Communications* thanks Hannah Kerner and the other, anonymous, reviewer(s) for their contribution to the peer review of this work. A peer review file is available.

