## [Peer Review File · Nature Communications]

Field-scale crop water consumption estimates reveal potential water savings in California agricultureREVIEWER COMMENTS

Reviewer #1 (Remarks to the Author):

This study reveals how much of the evapotranspiration (ET) in agricultural fields is due to the specific crop being grown there, as opposed to the total ET which includes natural ET that would occur without crop growth (fallow field) in addition to the agricultural ET. This is made possible by using an ML model to predict fallow/natural ET and subtracting the natural ET from the observed total ET provided by OpenET (derived from satellite observations). The agricultural ET maps are then used to evaluate the reduction in ET (reduction in actual water use) that could be obtained from switching crop types, changing farming practices, or fallowing fields. The study finds that all three scenarios have similar total reductions, but the fallowing scenario required land use changes in only 5% of fields while the other two scenarios impacted 50% of fields. These results can be used by decision-makers/policymakers to determine the most effective (in terms of cost, environmental, and social factors) strategies for reducing water use.

Overall I found the manuscript to be well written, easy to understand, and to have addressed most of the potential limitations of the analysis. There are only a few points that I had questions about:

1. One of the water-saving scenarios discussed is described as changing farming practices, which involves keeping the same spatial allocation of crops but changing the highest consumers to the median consumption level. While it cannot be stated for certain that the variations in agricultural ET within crops are due to farming practices, the authors showed that only a small percentage of the variations could be explained by other factors and thus is most likely due to farming practices. However, there was no discussion of what those farming practices actually are, and I did not see any analysis to show that any of the variation was due to farming practices. Is this due to a lack of data about farming practices? It's unclear without this discussion what would actually be involved in changing farming practices (particularly from a policy/economic perspective – would this be a major change or small change for a farmer?).
2. Are there errors associated with the OpenET observations that should be propagated when used in Eqns 1 and 4?
3. Lines 313-314 state that the analysis could only be conducted for years 2016, 2018, and 2019 because the year was used as a covariate in the ML model to predict natural ET. How sensitive are the model predictions to this variable? It seems like this could be a limiting factor for applying the same methodology to other locations/years since data will not always be available for every year, so it would be helpful to know how important that variable is to make accurate predictions.

Minor comments

- Line 208 has a typo: "irritation" instead of "irrigation"
- The split percentage for the fallow ET training dataset is described, but I did not see the total number of samples. Can this be added?

Reviewer #2 (Remarks to the Author):

Review: Field-scale crop water consumption estimates reveal potential water saving in California agriculture

This paper led by Boser et al. attempts to separate naturally occurring ET (ET_{nat}) and agriculture ET (ET_{ag}) in the Central Valley using remotely sensed ET and machine learning model. The estimates of ET_{ag} are then being used to understand patterns and derive scenarios for water savings, with an emphasis on land management (fallowing and alternative crops) and irrigation efficiency scenarios. The idea of separating the naturally occurring ET is novel, but the methodological approach does not seem robust. The estimates of irrigation efficiency seems flawed, and it fails to acknowledge/discuss the irrigation efficiency paradox (i.e., higher efficiency != more water saving, see:

<https://www.science.org/doi/10.1126/science.aat9314>). The pathway of applying water-saving land management scenarios is interesting and can be applied to other areas with minor manipulation. However, the paper failed to address the opportunity cost, water saving is only one of the many goals that farmers are trying to achieve. Below, I provide my detailed comments:

1) Separating the naturally occurring ET is the main crux of the paper. However, I am unclear how this was achieved. The paper talks about ML model, predictors used, and splitting the data between training, validation, and testing. What was the source of natural ET for building the model? I am guessing, based off of the description of fallow land identification, it was derived from OpenET. If this is true then, I am not confident that OpenET can accurately represent natural ET on fallow land. In my understanding OpenET was not validated for this environment and an over-prediction is very likely and quite apparent in Fig. 2. Natural ET for Pasture and Rice is closely following Agriculture ET and in many counties (San Joaquin and many neighboring counties) natural ET is more than agriculture ET (Fig. 1). 46% ET being contributed from naturally occurring ET on fallow land with no residual crop and in a Mediterranean climate warrant more ground truthing. What is the uncertainty associated with using PET data at such a coarse 0.1 degree resolution?

2) Concluding that crop switching and/or land fallowing as a scenario for saving water is not novel, PPIC and others have estimated potential water savings. In fact, PPIC extended their analysis to market and other considerations for the growers (<https://www.ppic.org/publication/exploring-the-potential-for-water-limited-agriculture-in-the-san-joaquin-valley/> , <https://www.sciencedirect.com/science/article/pii/S0048969722070632>). Authors should consider more comprehensive analysis of cost and benefits of different alternatives.

3) The calculation of irrigation efficiency seems flawed. First, I do not know the details of the USGS delivery data and whether it accounts for water moving in and out of counties. Looking at the map (Fig. 4), it is hard to comprehend that Mariposa and Madera counties have higher irrigation than Merced and Fresno. San Francisco and Marine counties also jump out. The north south valley split in the irrigation efficiency (Fig. 4, E8) suggests me that USGS data is likely incomplete and this pattern may very well driven by the local and state/federal water deliveries. Or it could be an artifact of your analysis time mismatch. Also, the comparison between theoretical and empirical irrigation efficiency is unfair since theoretical efficiencies do not account for conveyance losses.

4) Discussion section is very superficial and can be improved by referencing prior work on irrigation efficiency paradox, complexities with land fallowing and crop switching, market pressure, water right, land retirement for other beneficial use like solar, wildlife habitat, and even FloodMAR.

5) Lastly, I am intrigued by your use of the word empirical vs. modeled. Example: Agriculture ET derived from CalSIMETAW is being referred as modeled but same data from OpenET is empirical. Even for the sake of clarity, it would be better to refer the data by source.

Specific Comments:

a) More information should be provided on how various variables are influencing ET_{nat} prediction using ML model. It was briefly mentioned in discussion section (line 240-242) but is not well-defined.

b) The spatial scale of all analyses differs from each other. ET_{nat} was carried out at 70m resolution, variation in ET at Valley, water saving potential at groundwater sub-basin and irrigation efficiency at county level. More information on how spatial aggregation was done for the ET components from 70m to county/sub-basins scales and reasons for choosing these scales will be helpful.

c) Add details on crop groupings shown in Fig. 2 in the supplementary material.

d) Fig. A4 can be represented better with contrasting colors, hard to see the overlap. Also, plots with combination of bar and box plots can be improved with different color choices and adding gridlines.

e) The title seems misleading as the field-scale estimates are based of remotely sensed and other data of different scales, aggregated for the suggested water saving strategies at county/groundwater sub-basin scales. You may want to consider revising it.

f) Some sentences are lengthy and can be further simplified in the Introduction section. For example: lines 44-47, 68-71, 51-53.

g) Given how the ML model was developed, should you be still calling the ET over fallow land as natural? Don't you think fallow ET would be more appropriate?

Line-specific comments:

L46: evapotranspiration is “both” evaporation and transpiration that often occur simultaneously.

L111: “R2”

L124: More information on how 34% was calculated will be helpful for readers.

L125: “water intensive crops”

L128: reference to “(Figure A1)” seems wrong.

L207-208: You can write that “... 30% deficit irrigation for some crops (wheat, maize, cotton etc.)” as the referred studies are only for limited crops.

L211-214: The point made here makes sense, but not sure how relevant is the precipitation factor for Central valley agriculture as the peak demand and rainfall occurs are different times of the year.

Appendix C: Provide citation for CalSIMETAW

Reviewer #3 (Remarks to the Author):

Review of manuscript titled: “Field-scale crop water consumption estimates reveal potential water savings in California agriculture”

Summary: Authors present a study where through satellite remote sensing and machine learning they can differentiate between naturally occurring evapotranspiration (ET) and agricultural ET (the increase in ET induced by irrigation) to better quantify agricultural water use in the state of California; and by doing so offer potential solutions to decrease agricultural water use. This study is timely and relevant and takes advantage of the emerging technologies and approaches now available to earth system scientists. The scope of the manuscript is ambitious, but the authors have done a good job of explaining the steps and approaches/assumptions taken. I do believe the approach taken is novel (the subtraction of the naturally occurring ET) but am not entirely convinced of its validity. Other studies have quantified crop water use over the state using ET (Wong et al., 2021; Schauer et al., 2019; among others), but are not discussed in the manuscript. This may simply be due to the writing constraints of Nature Communications, but I still believe it is important to bring up.

Overall, I think this is a wonderful and important study. But I do have some general concerns that I think should be addressed prior to publication.

Concern #1. This is a novel approach. Quantifying agricultural ET by determining the ET that would be naturally occurring in that location should the crop (and applied irrigation) not be present and subtracting it from the actual ET observed via OpenET. That said, I’m hesitant to agree that the definition of agricultural ET is this difference. My hesitation stems from the results stating that, on average/in general, 46% of total ET is defined as naturally occurring and 53% of total ET is defined as agricultural ET. Intuitively this seems very high for ‘naturally occurring’. If I were in an almond orchard in September, I would not expect that 46% of the total ET coming off the field would be naturally occurring and only 53% would be because of the almond tree and the applied irrigation. I know you have broken these percentages down into crop categories, but they still seem high. I’m also aware that there are likely some time dependency factors that play a role. That said, I’m curious if you have any citations confirming this is a viable assumption or if you have ground data that can be used to validate your assumption. Also, because the idea behind the study is to quantify agricultural ET, should there be more of a month-by-month analysis or growing season analysis? I think it is important prior to scaling an approach like this to the entire state and making claims about water use that a proof of concept be completed, either on the ground or found in the literature.

Concern #2. I apologize if I missed this in the manuscript, or if it is common knowledge, but could you please provide a thorough description of “diversion” that is used throughout the manuscript in reference to water diversions and irrigation. My interpretation is that diversion is water found/stored in one location (i.e., county) and “diverted” to another location (i.e., another county). This interpretation makes sense when looking at Fig. 4 (left column) where we find higher water diversions for irrigation in the north (the Sacramento River Valley and north). Water in the northern portion of the state is famously (or infamously, depending on where in the state you are) moved further south for agricultural irrigation. In my mind this makes the southern counties much less efficient (in terms of

irrigation efficiency) compared to the north. However, the opposite is presented, with the southern counties reporting higher irrigation efficiencies. Could you please offer more context to this Figure and the overall implications made from the associated analysis?

Concern #3. Different parts of the manuscript discuss the within crop variability in ET as well as statements alluding to how agricultural water applications can be significantly decreased (from 95 percentile to 25 percentile) without changing the landcover. First, a likely reason for ET variability within the same cropping system (particularly tree nuts) is because of the age of the orchard. These systems are perennials and exist for 15-25 years. It makes sense that an annual would present lower variation – its effectively the same crop every year. A pistachio orchard changes drastically from 2nd leaf to 20th leaf, with a considerable amount of increase to biomass and by proxy ET. In order to make the claim that water savings can occur without changing the crop, there needs to be an accounting of orchard age. Second, I do not think that scenario #2 in reducing agricultural ET is a viable scenario. Sure, deficit irrigation is possible and is well studied in certain cropping systems. In some, such as wine grapes, its required to produce a high-quality grape. But to suggest cutting back irrigation based on 'high' ET estimates to 'median' ET estimates without including age (again, perennials introduce some difficulty here), production goals of the commodity, or management style seems cavalier.

More specifics below:

Line 31: This first sentence is a bit confusing – are you saying the overexploitation of 'drought' and 'climate change' have led to it seems like there is simply some grammatical error tying the sentence together.

Line 43: "because a large share of irrigation water is returned to the system as runoff or recharge". I understand the statement and it is true, but because the study takes place in California, it is important to note the high efficiency of the irrigation in the state (drip or micro-sprinklers installed). In these systems, very little irrigated water is lost to runoff or deep percolation.

Line 46: Consider rephrasing in that evapotranspiration is not evaporated by the soil but rather from the soil to the atmosphere.

Line 51: This last sentence is a bit confusing to me. How does decreasing water diversions not affect food production? Because irrigation water does not contribute to crop growth? Maybe I'm missing something?

Line 112: Here you give error metrics in cm/year, but you specify they will be mm/year on line 285. I would suggest reporting values in mm/year as I've never seen ET reported in cm/year. It might also be worth mentioning that all growers use acre-in or acre-feet as their unit of choice. I would suggest converting your overall units (mm/day) to acre-in/acre-feet in maybe the conclusion/discussion so a broader readership can more quickly interpret your results.

Line 113: Your test set is made up of 2 km² regions – can you mention how many? Also, these regions represent fallowed land. Are all 2 km² regions completely fallowed land? I guess I'm asking if all fallowed pixels chosen are arbitrary 2 km² boundaries placed all over the central valley and not polygons outlining specific fallowed fields.

Line 137: See concern #3.

Line 149: See concern #3.

Line 208: Irrelevant and misleading. This 30% figure is from citations looking at wheat, maize, and cotton; all of which are annuals and all of which have no notable economic viability in California (combined, their economic output is only 2% of almonds and grapes combined).

Line 250: Sorry for likely not following, but isn't this 7% much lower than values specified earlier? Where is this coming from?

Response to reviewers: Field-scale crop water consumption estimates reveal potential water savings in California agriculture

November 25, 2023

We would like to thank all of the reviewers for their thoughtful comments which have guided us in the revision of this manuscript. This document provides a guide to the nature and location of the changes prompted by the comments. Below, we reproduce each reviewer comment italicized in gray and provide our replies in black. Excerpts from the paper are indented with the line numbers noted, with changes highlighted in red. Figures that appear in the main text or SI retain their original names and numbering. Additional figures and tables are prefixed with the letter “R” in order to distinguish them from figures in the paper.

1 Reviewer #1

This study reveals how much of the evapotranspiration (ET) in agricultural fields is due to the specific crop being grown there, as opposed to the total ET which includes natural ET that would occur without crop growth (fallow field) in addition to the agricultural ET. This is made possible by using an ML model to predict fallow/natural ET and subtracting the natural ET from the observed total ET provided by OpenET (derived from satellite observations). The agricultural ET maps are then used to evaluate the reduction in ET (reduction in actual water use) that could be obtained from switching crop types, changing farming practices, or fallowing fields. The study finds that all three scenarios have similar total reductions, but the fallowing scenario required land use changes in only 5% of fields while the other two scenarios impacted 50% of fields. These results can be used by decision-makers/policymakers to determine the most effective (in terms of cost, environmental, and social factors) strategies for reducing water use.

Overall I found the manuscript to be well written, easy to understand, and to have addressed most of the potential limitations of the analysis. There are only a few points that I had questions about:

1.1 Discussion of farming practices

1. *One of the water-saving scenarios discussed is described as changing farming practices, which involves keeping the same spatial allocation of crops but changing the highest consumers to the median consumption level. While it cannot be stated for certain that the variations in agricultural ET within crops are due to farming practices, the authors showed that only a small percentage of the variations could be explained by other factors and thus is most likely due to farming practices. However, there was no discussion of what those farming practices actually are, and I did not see any analysis to show that any of the variation was due to farming practices. Is this due to a lack of data about farming practices? It's unclear without this discussion what would actually be involved in changing farming practices (particularly from a policy/economic perspective – would this be a major change or small change for a farmer?).*

Thank you for your comment, which has prompted us to revisit our discussion of the farming practices scenario. As you suspected, the reason we do not conduct further analyses to identify the water saving potential of specific farming practices is because there are no field-level data available in California that would allow for such an analysis. However, there is published research investigating the water-saving potential of individual farming practices in particular locations. We have updated the paper to include a more detailed discussion of this prior literature and its implications for our findings.

We have made three specific changes in response to this comment. First, we have expanded our discussion to include previous literature on water saving farming practices. We cite practices such as mulching, conservation tillage, deficit irrigation, and improved irrigation scheduling and technologies. Second, we touch on the economic and policy considerations involved in farm-level management changes by highlighting the relative cost-effectiveness of these practices compared to other water-saving strategies. Finally, we underscore the importance of assembling and sharing data on such farming practices for future research, as it would enable a more comprehensive assessment of the water-saving potential of various farming practices.

These changes are reflected in paragraph 3 of the discussion (line 243):

[...] Even after adjusting for variability attributable to field characteristics, orchard age, and climate, we find that these variations could translate to substantial water savings without requiring a change in crop type. Unfortunately, a lack of high-resolution data on field-scale farming practices and yields inhibits further analysis of specific practices driving agricultural ET variation and their economic implications. However, prior literature from experimental plots or particular locations suggests that mulching (Kader et al., 2019), conservation tillage (Mitchell et al., 2012), deficit irrigation (English and Raja, 1996; Goldhamer, 1999; Rudnick et al., 2019), and improved irrigation scheduling and technologies (Jovanovic et al., 2020) all have potential to limit agricultural ET. These practices may prove advantageous relative to costly strategies like crop-switching or fallowing, though more detailed cost-benefit analyses are necessary to determine the suitability of various interventions in specific contexts (Mitchell et al., 2016). Such water saving farming practices are not mentioned in the plans drafted by SGMA water managers (Bruno et al., 2022). This is possibly due to a lack of conclusive research on the

potential of such strategies to effectively decrease agricultural ET without significant effects on yield or operation cost. Spatial data on the use of these different practices would allow researchers to take full advantage of our high-resolution agricultural ET estimates and study their water-saving and economic benefits.

1.2 Error propagation

2. Are there errors associated with the OpenET observations that should be propagated when used in Eqns 1 and 4?

We appreciate your query regarding the propagation of errors from OpenET and take this opportunity to clarify this aspect of our methodology and enhance the rigor of our analysis. We note that because eq. 1, $ET_{ag} = ET_{tot} - ET_{nat}$, serves as a definition of what we consider a theoretical, “true” agricultural ET, we do not introduce error terms at that stage. However, errors certainly do affect our empirical estimates of agricultural ET, which could for example propagate into eq. 4, $AdjustedET_{p,y} = \beta_{p,y} \times Group_{p,y} + \epsilon_{p,y}$. In our revision, we have added both a conceptual and empirical investigation of error propagation, which shows how errors both from OpenET and from our machine learning model may influence our downstream estimates of agricultural ET. We conclude that empirically, these errors are insufficient to influence key findings of the paper.

We detail our new analysis on this topic in Supplementary Note 4 copied below, but summarize it briefly here. Several error terms are introduced into the agricultural ET estimates: two from error in OpenET and one from error in our machine learning model. These errors have the potential to introduce bias into our estimates of agricultural ET and to inflate estimated variance over space and/or time in agricultural ET. Using OpenET’s Intercomparison and Accuracy report (OpenET, 2021) and our own validation of the machine learning model (Supplementary Fig. 1, 2, 3, and 4), we find that all of these errors are unbiased (mean zero). Because they are unbiased, they are captured in the random error term in the regression equations 3 and 4. We consequently do not adjust these equations in the manuscript.

However, error still artificially inflates variance in agricultural ET. This impacts our analysis of the water saving potential of farmer’s practices and fallowing, because variations in estimated agricultural ET across space is used to investigate possible water saving. We now show that the variance of our estimated agricultural ET is equal to the true variance in agricultural ET plus the variance of the error we accrue. However, we estimate that variance of the error terms is only 11% of the total variance we find across our agricultural ET estimates. Therefore, the vast majority of the variation we retrieve is in fact signal rather than noise. We include this decomposition of error in new Supplementary Note 4 (SI line 76) and we additionally note this finding in the discussion in the main text (line 298).

Supplementary Note 4: The effect of OpenET and machine learning model error on agricultural ET estimates

We define agricultural ET as the difference between total ET and naturally-occurring ET, $ET_{ag} = ET_{tot} - ET_{nat}$ (Main text eq. 1), where naturally-occurring ET is the counterfactual ET that would occur naturally, were the same land fallow. However, we calculate agricultural ET using estimates of total ET from OpenET and modeled naturally-occurring ET trained on OpenET data (Supplementary eq. 1). Since the OpenET data and the model we use to estimate naturally-occurring ET have errors associated with them, our estimates of agricultural ET can be described as the true agricultural ET plus several error terms (Supplementary eq. 4), which we derive as follows:

$$ET_{ag}^* = E\hat{T}_{tot} - E\tilde{T}_{nat} \quad (1)$$

where ET_{ag}^* denotes our estimate of agricultural ET, $E\hat{T}_{tot}$ is OpenET's estimate of ET_{tot} and $E\tilde{T}_{nat}$ is our estimate of the naturally-occurring ET, predicted by our machine learning (ML) model. The hat notation indicates estimates inclusive of OpenET error, while the tilde indicates estimates inclusive of ML model error.

Because our naturally-occurring ET ML model is trained on and therefore predicts OpenET values over fallow lands, we can rewrite the same expression as:

$$ET_{ag}^* = E\hat{T}_{tot} - (E\hat{T}_{nat} + \epsilon_{ML}) \quad (2)$$

where $E\hat{T}_{nat}$ is the (naturally-occurring) ET that OpenET would observe if the field were fallow, and ϵ_{ML} is the error from the ML model in predicting $E\hat{T}_{nat}$.

We can then again reorganize the same equation by separating the error from the OpenET observations. This reveals the relationship between the true ET_{ag} and our estimate, ET_{ag}^* :

$$ET_{ag}^* = ET_{tot} + \epsilon_{OpenET,ag} - (ET_{nat} + \epsilon_{OpenET,fal} + \epsilon_{ML}) \quad (3)$$

$$= ET_{ag} + \epsilon_{OpenET,ag} - \epsilon_{OpenET,fal} - \epsilon_{ML} \quad (4)$$

where ET_{ag} , ET_{tot} , and ET_{nat} are the true (unobservable) agricultural ET, total ET, and naturally-occurring ET, respectively, $\epsilon_{OpenET,ag}$ is the error associated with the OpenET estimate over the agricultural field and $\epsilon_{OpenET,fal}$ is the error associated with what the OpenET estimate would be if the land were fallow.

In our study, we seek to estimate mean ET_{ag} , be it over the entire Central Valley, for a certain crop type, or for a given county, we seek to calculate the expectation of ET_{ag} over a sample of grid cells. Using Supplementary eq. 5, we can show that our estimate of $\mathbb{E}[ET_{ag}^*]$ depends not only on $\mathbb{E}[ET_{ag}]$, but also the expectation of the error terms.

$$\mathbb{E}[ET_{ag}^*] = \mathbb{E}[ET_{ag}] + \mathbb{E}[\epsilon_{OpenET,ag}] - \mathbb{E}[\epsilon_{OpenET,fal}] - \mathbb{E}[\epsilon_{ML}] \quad (5)$$

We show in Supplementary Figures 1-4 that ϵ_{ML} is unbiased, that is, $\mathbb{E}[\epsilon_{ML}] = 0$. OpenET has also been shown to produce unbiased estimates over agricultural lands, so $\mathbb{E}[\epsilon_{OpenET,ag}] = 0$ (see Table 3 on page 6 of the OpenET Intercomparison and Accuracy report) (OpenET, 2021). We can also assume $\epsilon_{OpenET,fal}$ is unbiased because though OpenET has not specifically been evaluated over fallow fields, it produces unbiased estimates over natural shrublands and grasslands with ET values of similar magnitude to fallow fields in the Central Valley (OpenET, 2021). As a result, in expectation our estimates of agricultural ET are unbiased, and consequently produce unbiased regression coefficients.

$$\mathbb{E}[ET_{ag}^*] = \mathbb{E}[ET_{ag}] \quad (6)$$

While our estimates of agricultural ET are unbiased in expectation, the error terms do add variance that is not present in the true agricultural ET. Therefore, any analysis that assesses variation across pixels will reflect inflated variance relative to true agricultural ET. For example, this has implications for the farming practices and following scenarios we conduct (Fig. 3). However, here we estimate that error is responsible for only 11% of the variance in our annual estimates of agricultural ET, suggesting its influence over key results is limited. Specifically, if we assume that all error terms are independent, we have:

$$var(ET_{ag}^*) = var(ET_{ag}) + var(\epsilon_{OpenET,ag}) + var(\epsilon_{OpenET,fal}) + var(\epsilon_{ML}) \quad (7)$$

where $var(\cdot)$ denotes the variance.

Then, because variance is equal to mean squared error (MSE) when bias is 0, we have:

$$var(ET_{ag}^*) = var(ET_{ag}) + MSE(\epsilon_{OpenET,ag}) + MSE(\epsilon_{OpenET,fal}) + MSE(\epsilon_{ML}) \quad (8)$$

We calculate the MSE of our machine learning model using our test set, and OpenET provides root MSE values over croplands and shrublands (which we use to approximate error over fallow lands) (again, see Table 3 on page 6 of the OpenET Intercomparison and Accuracy report) (OpenET, 2021). We find that for estimates of yearly ET_{ag}^* , $MSE(\epsilon_{OpenET,ag}) + MSE(\epsilon_{OpenET,fal}) + MSE(\epsilon_{ML})$ represents only 11% of $var(ET_{ag}^*)$. This indicates that the vast majority of the variance we uncover is representative of true differences in water consumption across fields, as opposed to variability due to model errors.

We feel that this additional analysis greatly strengthens our work, clarifying its potential and its limitations. As such, we have added a passage in the limitations section of the discussion that highlights the main takeaways of this inquiry (line 298):

This study has some important limitations. Most notably, **due to a lack of data on farming practices**, it is difficult to ascertain whether the variation in within-crop agricultural ET that we estimate is indeed due to farming practices. **When conducting our management scenarios, we account for the effects of climate, soil quality, topography, and orchard age. We additionally conduct the scenarios at the level of small groundwater sub-basins rather than across the entire valley to account for any additional regional environmental differences we are unable to otherwise account for. However, variance stemming from error in our agricultural ET estimates or from mislabeled crop types could contribute to observed within-crop variation in agricultural ET.** We minimize error from mislabeled crop types by using the most accurate crop data available in California which boasts an accuracy of 97.6% (LandIQ, 2021). **Additionally, the OpenET ensemble model has been extensively validated (Melton et al., 2022; OpenET, 2021), and our machine learning model has an R^2 of .87. We estimate that these sources of error are responsible for only 11% of the variance in our yearly agricultural ET estimates (Supplementary Note 4). Nevertheless, the water-saving potentials we calculate for both the fallowing and farmers practice scenarios should be interpreted as upper bound estimates.**

1.3 Using ‘year’ as a covariate

3. Lines 313-314 state that the analysis could only be conducted for years 2016, 2018, and 2019 because the year was used as a covariate in the ML model to predict natural ET. How sensitive are the model predictions to this variable? It seems like this could be a limiting factor for applying the same methodology to other locations per years since data will not always be available for every year, so it would be helpful to know how important that variable is to make accurate predictions.

Thank you for bringing to our attention the potential limitation imposed by using ‘year’ as a covariate in our model. To address this, we have performed an analysis of feature importance within our model, which included assessing the impact of the ‘year’ variable on the model’s predictions. Our findings suggest that the ‘year’ is not among one of the most critical predictors in the context of the overall model performance, contributing only 3% of the model’s predictive capability (Supplementary Fig. 17).

Supplementary Fig. 17. The contribution of each variable included in the gradient boosting model on its predictions of naturally-occurring ET.

In light of this, we believe that our methodology holds the flexibility to be adjusted for applications where ‘year’ data may be incomplete or entirely unavailable. We have included this insight in the methods section (line 379).

However, the inclusion of an indicator variable for each year in our model limits our analysis to 2016, 2018, and 2019, as these are the only years for which we have available land cover data. Nevertheless, given that ‘year’ contributes a mere 3% to our model’s predictive capability (see Supplementary Fig. 17), it may be considered non-essential for future research in this area.

We hope that these changes will provide a comprehensive view of the variable importance and offer additional insights into the robustness and adaptability of our method.

1.4 Minor comments

Line 208 has a typo: “irritation” instead of “irrigation”

Thank you for your attention to detail; we have fixed this misspelling.

The split percentage for the fallow ET training dataset is described, but I did not see the total number of samples. Can this be added?

Thank you for pointing out this oversight. We have added the following statement in the methods section (line 396).

To validate our naturally-occurring ET model, we split our dataset, reserving 60% for training, 10% for validation, and 30% for testing. In order to ensure that nearby and therefore very similar pixels are not present across multiple splits, we group our splits by 2 km² squares, four times the size of a large agricultural field in the Central Valley. The entire dataset is made up of over 16 million pixels populating 8,180 distinct 2 km² regions. Nearly 10 million pixels and 4,908 2 km² clusters are contained in the training split, and 2,454 clusters are reserved for testing.

2 Referee #2

Review: Field-scale crop water consumption estimates reveal potential water saving in California agriculture

This paper led by Boser et al. attempts to separate naturally-occurring ET (ET_{nat}) and agriculture ET (ET_{ag}) in the Central Valley using remotely sensed ET and machine learning model. The estimates of ET_{ag} are then being used to understand patterns and derive scenarios for water savings, with an emphasis on land management (fallowing and alternative crops) and irrigation efficiency scenarios. The idea of separating the naturally-occurring ET is novel, but the methodological approach does not seem robust. The estimates of irrigation efficiency seems flawed, and it fails to acknowledge/discuss the irrigation efficiency paradox (i.e., higher efficiency != more water saving, see: <https://www.science.org/doi/10.1126/science.aat9314>). The pathway of applying water-saving land management scenarios is interesting and can be applied to other areas with minor manipulation. However, the paper failed to address the opportunity cost, water saving is only one of the many goals that farmers are trying to achieve. Below, I provide my detailed comments:

Thank you for your valuable feedback, which has been instrumental in enhancing our paper. Regarding your overarching observations, we have made the following amendments, which we describe in greater detail below:

1. We conducted a comprehensive analysis of potential errors in our methodology for estimating agricultural ET and confirmed the robustness of our approach. Details of this analysis have been added to the supplementary information and are now referenced in the main text for further clarity.
2. In response to your concerns about our methodology for estimating irrigation efficiency, we have refined our approach for greater clarity. The manuscript additionally now includes an expanded discussion on the irrigation efficiency paradox, offering a more nuanced understanding of this complex issue.
3. We have enriched our study by incorporating a discussion of the opportunity costs associated with various water-saving strategies, as explored in prior literature. This addition aims to provide a more comprehensive perspective on the economic implications of these strategies.

2.1 Errors associated with OpenET

1) Separating the naturally-occurring ET is the main crux of the paper. However, I am unclear how this was achieved. The paper talks about ML model, predictors used, and splitting the data between training, validation, and testing. What was the source of natural ET for building the model? I am guessing, based off of the description of fallow land identification, it was derived from OpenET. If this is true then, I am not confident that OpenET can accurately represent natural ET on fallow land. In my understanding OpenET was not validated for this environment and an over-prediction is very likely and quite apparent in Fig. 2.

Natural ET for Pasture and Rice is closely following Agriculture ET and in many counties (San Joaquin and many neighboring counties) natural ET is more than agriculture ET (Fig. 1). 46% ET being contributed from naturally-occurring ET on fallow land with no residual crop and in a Mediterranean climate warrant more ground truthing. What is the uncertainty associated with using PET data at such a coarse 0.1 degree resolution?

Thank you for your thoughtful critique focusing on the critical aspect of separating naturally-occurring ET in our paper. As we describe in detail below, your comments have prompted us to (1) refine our explanation of how we achieve this separation and (2) conduct a comprehensive analysis of potential errors, particularly those associated with our use of OpenET data. This analysis has been instrumental in enhancing the robustness of our methodology and ensuring the accuracy of our findings. Lastly, we address your concern surrounding the use of coarse PET data.

2.1.1 Clarifying our methods

We apologize for the lack of clarity in our methods. You are correct that we used OpenET estimates over fallow lands to train the model predicting naturally-occurring ET. In order to improve clarity surrounding this point, we revise the introduction where we first introduce our approach (line 95):

Here, we develop a novel framework for measuring agricultural ET at sub-field scales, using remote sensing to determine total ET and machine learning to estimate naturally-occurring ET. First, we retrieve satellite-based remotely sensed total ET estimates from the 30m, monthly OpenET ensemble data (Melton et al., 2022) available in the western United States starting in 2016. Second, since we can **use these same OpenET estimates to** directly observe naturally-occurring ET over fallow lands, we train a gradient boosting algorithm to predict ET over fallow lands, using information on topography, soil quality, climate, and spatial **and temporal** coordinates. We then use the model to retrieve naturally-occurring ET over all active agricultural fields, which we subtract from the remotely-sensed ET to calculate agricultural ET.

Additionally, we add a sentence at the beginning of the results section to further clarify this point (line 120):

We calculate agricultural ET by retrieving the total ET observed over agricultural areas and subtracting naturally-occurring ET (Fig. 1). **While total ET estimates are retrieved from OpenET, we simulate naturally-occurring ET by training a gradient boosting regressor to predict the ET observed by OpenET over fallow lands.**

2.1.2 Errors associated with our estimates of naturally-occurring ET

We are committed to ensuring the accuracy of our findings and clarity of our work. To ensure the adequacy of our naturally-occurring ET estimates, we follow several steps including the consideration of potential biases arising from OpenET data.

1. **Investigation of OpenET and ML model errors.** We investigate potential errors associated with OpenET and our machine learning model, and find no reason to suspect biased estimates arising from either. We undertake an analysis fully detailing the role of both in our estimates of agricultural ET (Supplementary Note 4).
2. **Additional analyses.** To further ensure the adequacy of our estimates of naturally-occurring ET, we additionally undertake an analysis of land cover data accuracy (Supplementary Note 6), conduct a comparison with other estimates, and consider the robustness of our findings against hidden biases.
3. **Clarification of annual ET values.** We recognize that our naturally-occurring ET values can seem high since we report annual values as opposed to ones confined to the growing season. To stave off confusion, we edit the manuscript to underscore this fact.

Investigation of OpenET and ML model errors. While it is true that OpenET’s validation has been focused on croplands and has not been validated over fallow lands specifically, it has been validated over natural lands such as grasslands and shrublands. According to OpenET’s Intercomparison and Accuracy Assessment Report, croplands, grasslands, and shrublands, all which may help represent the kinds of errors we might expect over fallow lands, exhibit no statistically significant biases, and therefore do not suggest that OpenET is positively biased over fallow lands (Table R 1) (OpenET, 2021). Shrublands may especially be considered as a good proxy for the fallow lands of the Central Valley since the average ET of the shrublands used in the OpenET validation, 31.07 mm/month, is similar to what OpenET predicts over fallow lands in the Central Valley (mean = 36.5 mm/month).

To improve transparency surrounding the effect of errors inherent to our method, we have added a section in the supplementary information detailing the role of error in our analysis (Supplementary Note 4, SI line 76).

Supplementary Note 4: The effect of OpenET and machine learning model error on agricultural ET estimates

We define agricultural ET as the difference between total ET and naturally-occurring ET, $ET_{ag} = ET_{tot} - ET_{nat}$ (Main text eq. 1), where naturally-occurring ET is the counterfactual ET that would occur naturally, were the same land fallow. However, we calculate agricultural ET using estimates of total ET from OpenET and modeled naturally-occurring ET trained on OpenET data (Supplementary eq. 1). Since the OpenET data and the model we use to estimate naturally-occurring ET have errors associated with them, our estimates of agricultural ET can be described as the true

Land cover type	Statistic	Ensemble	Range
Croplands 45 sites, N = 1682 months Mean station ET = 93.68 (mm)	Slope	0.95	0.86 - 1.04
	MBE (mm)	-3.64 (-3.9%)	-13.77 - 5.16 (-14.7% - 5.5%)
	MAE (mm)	155.66 (16.6%)	17.96 - 22.92 (19.2% - 24.5%)
	RMSE (mm)	195.17	23.43 - 25.89 (25% - 27.7%)
	R-squared	0.95	0.89 - 0.93
Grasslands 20 sites, N = 672 months Mean station ET = 42.56 (mm)	Slope	1.03	0.73 - 1.28
	MBE (mm)	-1.23 (-2.9%)	-11.88 - 9.73 (-27.9% - 22.9%)
	MAE (mm)	19.45 (15.4%)	20.17 - 28.1 (47.4% - 66.0%)
	RMSE (mm)	24.12 (56.7%)	24.62 - 35.96 (57.8% - 84.5%)
	R-squared	0.75	0.54 - 0.8
Shrublands 24 sites, N = 681 months Mean station ET = 31.07 (mm)	Slope	0.95	0.65 - 1.34
	MBE (mm)	2.89 (9.3%)	-5.2 - 18.7 (-16.7% - 41.4%)
	MAE (mm)	15.68 (50.5%)	17.45 - 22.64 (56.2% - 72.9%)
	RMSE (mm)	19.96 (64.2%)	20.9 - 29.18 (67.3% - 93.9%)
	R-squared	0.66	0.3 - 0.69

Table R 1. OpenET monthly performance over croplands, grasslands, and shrublands. MBE denotes mean bias error, and none of the land cover types show statistically significant MBE. Reproduced from the OpenET Intercomparison and Accuracy Assessment Report, Table 3 (pg.6).

agricultural ET plus several error terms (Supplementary eq. 4), which we derive as follows:

$$ET_{ag}^* = E\hat{T}_{tot} - E\tilde{T}_{nat} \quad (1)$$

where ET_{ag}^* denotes our estimate of agricultural ET, $E\hat{T}_{tot}$ is OpenET's estimate of ET_{tot} and $E\tilde{T}_{nat}$ is our estimate of the naturally-occurring ET, predicted by our machine learning (ML) model. The hat notation indicates estimates inclusive of OpenET error, while the tilde indicates estimates inclusive of ML model error.

Because our naturally-occurring ET ML model is trained on and therefore predicts OpenET values over fallow lands, we can rewrite the same expression as:

$$ET_{ag}^* = E\hat{T}_{tot} - (E\hat{T}_{nat} + \epsilon_{ML}) \quad (2)$$

where $E\hat{T}_{nat}$ is the (naturally-occurring) ET that OpenET would observe if the field were fallow, and ϵ_{ML} is the error from the ML model in predicting $E\hat{T}_{nat}$.

We can then again reorganize the same equation by separating the error from the OpenET observations. This reveals the relationship between the true ET_{ag} and our estimate, ET_{ag}^* :

$$ET_{ag}^* = ET_{tot} + \epsilon_{OpenET,ag} - (ET_{nat} + \epsilon_{OpenET,fal} + \epsilon_{ML}) \quad (3)$$

$$= ET_{ag} + \epsilon_{OpenET,ag} - \epsilon_{OpenET,fal} - \epsilon_{ML} \quad (4)$$

where ET_{ag} , ET_{tot} , and ET_{nat} are the true (unobservable) agricultural ET, total ET, and naturally-occurring ET, respectively, $\epsilon_{OpenET,ag}$ is the error associated with the OpenET estimate over the agricultural field and $\epsilon_{OpenET,fal}$ is the error associated with what the OpenET estimate would be if the land were fallow.

In our study, we seek to estimate mean ET_{ag} , be it over the entire Central Valley, for a certain crop type, or for a given county, we seek to calculate the expectation of ET_{ag} over a sample of grid cells. Using Supplementary eq. 5, we can show that our estimate of $\mathbb{E}[ET_{ag}^*]$ depends not only on $\mathbb{E}[ET_{ag}]$, but also the expectation of the error terms.

$$\mathbb{E}[ET_{ag}^*] = \mathbb{E}[ET_{ag}] + \mathbb{E}[\epsilon_{OpenET,ag}] - \mathbb{E}[\epsilon_{OpenET,fal}] - \mathbb{E}[\epsilon_{ML}] \quad (5)$$

We show in Supplementary Figures 1-4 that ϵ_{ML} is unbiased, that is, $\mathbb{E}[\epsilon_{ML}] = 0$. OpenET has also been shown to produce unbiased estimates over agricultural lands, so $\mathbb{E}[\epsilon_{OpenET,ag}] = 0$ (see Table 3 on page 6 of the OpenET Intercomparison and Accuracy report) (OpenET, 2021). We can also assume $\epsilon_{OpenET,fal}$ is unbiased because though OpenET has not specifically been evaluated over fallow fields, it produces unbiased estimates over natural shrublands and grasslands with ET values of similar magnitude to fallow fields in the Central Valley (OpenET, 2021). As a result, in expectation our estimates of agricultural ET are unbiased, and consequently produce unbiased regression coefficients.

$$\mathbb{E}[ET_{ag}^*] = \mathbb{E}[ET_{ag}] \quad (6)$$

While our estimates of agricultural ET are unbiased in expectation, the error terms do add variance that is not present in the true agricultural ET. Therefore, any analysis that assesses variation across pixels will reflect inflated variance relative to true agricultural ET. For example, this has implications for the farming practices and fallowing scenarios we conduct (Fig. 3). However, here we estimate that error is responsible for only 11% of the variance in our annual estimates of agricultural ET, suggesting its influence over key results is limited. Specifically, if we assume that all error terms are independent, we have:

$$var(ET_{ag}^*) = var(ET_{ag}) + var(\epsilon_{OpenET,ag}) + var(\epsilon_{OpenET,fal}) + var(\epsilon_{ML}) \quad (7)$$

where $var(\cdot)$ denotes the variance.

Then, because variance is equal to mean squared error (MSE) when bias is 0, we have:

$$var(ET_{ag}^*) = var(ET_{ag}) + MSE(\epsilon_{OpenET,ag}) + MSE(\epsilon_{OpenET,fal}) + MSE(\epsilon_{ML}) \quad (8)$$

We calculate the MSE of our machine learning model using our test set, and OpenET provides root MSE values over croplands and shrublands (which we use to approximate

error over fallow lands) (again, see Table 3 on page 6 of the OpenET Intercomparison and Accuracy report) (OpenET, 2021). We find that for estimates of yearly ET_{ag}^* , $MSE(\epsilon_{OpenET,ag}) + MSE(\epsilon_{OpenET,fal}) + MSE(\epsilon_{ML})$ represents only 11% of $var(ET_{ag}^*)$. This indicates that the vast majority of the variance we uncover is representative of true differences in water consumption across fields, as opposed to variability due to model errors.

Analysis of land cover data accuracy. The only remaining source of error which could bias our estimates of agricultural ET would be if the land cover data we use to identify fallow lands were imprecise. If this were the case, we would likely be training our naturally-occurring ET machine learning model on agricultural fields. This would lead to an overestimation of naturally-occurring ET and therefore an underestimation of agricultural ET.

To rule out this potential issue, we test the effect of train a model exclusively on land that was marked fallow not only by the DWR dataset we use for our main results, but also by the Cropland Data Layer (CDL). We find that the results do not change significantly with this more stringent criterion, and report these findings in our supplementary information (Supplementary Note 6, Supplementary Fig. 13, 14, and 15) (SI line 142).

Supplementary Note 6: Results when using training data also marked fallow by the Cropland Data Layer (CDL)

When training the naturally-occurring ET model, we rely on the California Department of Water Resources (DWR) LandIQ crop type dataset due to its high accuracy. Here, we evaluate the effect of instead training the data only on fallow fields that are marked as fallow by both the DWR dataset and the Cropland Data Layer (CDL). Overall, this change does not affect our estimates of naturally-occurring ET: while we estimate 438 mm per year (394.4-481.7 95% CI) in the main text, we estimate 423.9 mm per year (369-478.7 95% CI) here. All general trends and conclusions are robust to this change (Supplementary Fig. 14, Fig. 15, Fig. 16).

Supplementary Fig. 14. Variations in agricultural ET across and within crop groups using training data also marked fallow by the Cropland Data Layer. This is the equivalent of Fig. 2 in the main text, but calculated using a naturally-occurring ET model trained on data also marked fallow by the Cropland Data Layer.

Supplementary Fig. 15. The percent reduction in agricultural ET driven by various management scenarios using training data also marked fallow by the Cropland Data Layer. This is the equivalent of Fig. 3 in the main text, but calculated using a naturally-occurring ET model trained on data also marked fallow by the Cropland Data Layer.

Supplementary Fig. 16. Irrigation efficiency across the counties of the Central Valley using training data also marked fallow by the Cropland Data Layer. This is the equivalent of Fig. 4 in the main text, but calculated using a naturally-occurring ET model trained on data also marked fallow by the Cropland Data Layer.

Comparison with other estimates. We additionally note that the numbers we uncover for naturally-occurring ET are within reason when compared to precipitation within the Central Valley: while CalSIMETAW reports an average precipitation of 554 mm/year, our naturally-occurring ET estimates amount to 438 mm/year (394.4-481.7 95% CI) (Orang et al., 2013). Additionally, we find that 438 mm/year falls within the range of values found in the literature studying natural systems in Mediterranean climates, which typically report values between 300 and 480 mm/year (Aires et al., 2008; Baldocchi et al., 2004; Duell, 1988).

Robustness of findings against hidden biases. These additional analyses do not suggest that OpenET or other potential sources of error would be inflating our estimates of naturally-occurring ET. However, out of an abundance of caution, we consider how our findings would be affected if our estimates of naturally-occurring ET were nevertheless inflated, causing artificially low estimates of agricultural ET. Because our main findings focus more on relative differences in agricultural ET, the patterns we uncover would remain unchanged. For example, the ranking in water intensity of crops would not be affected by a uniform underestimation of agricultural ET. Our scenarios all report water savings in percent, and would also remain unchanged. Our estimates of irrigation efficiency would increase, but our main finding surrounding the pronounced north to south gradient would not be affected.

Clarification of annual ET values. While we do not identify any potential sources of bias from our methods, we recognize that our estimates of naturally-occurring ET can seem high since they are aggregated over the entire year and include naturally-occurring ET that occurs at times where the water is unavailable to crops. As you note, this is especially problematic in Fig. 2, where the naturally-occurring ET makes up a large fraction of total ET across crop types. In order to stave off confusion, we edit the caption of this figure to clarify that not all naturally-occurring ET necessarily translates to crop growth. We additionally remove the statement on the proportion of annual total ET attributable to naturally-occurring ET to avoid readers interpreting it as the fraction of water used by crops that is naturally-occurring.

Fig. 2. Variations in annual agricultural ET across and within crop groups. Mean agricultural ET by crop group (green fill and 95% CI) is the average difference between total ET (black outline) and naturally-occurring ET (brown fill). All measures are summed across the year, leading to naturally-occurring and total ET estimates that include water consumption occurring outside of the growing season. While we find significant differences in mean agricultural ET across crop groups, the gray box plots also show a broad spread in agricultural ET within crop groups (box plots show 0.5, 0.25, 0.5, 0.75, and 0.95 quantiles).

2.1.3 The effect of using a coarse PET product

Regarding your query about the use of coarse PET data in predicting naturally-occurring ET, we apologize for the lack of clarity surrounding the methodological decisions behind our model. We explain these choices below and additionally incorporate them into our manuscript to improve clarity for our readers.

Our model is composed of multiple predictors, including latitude and longitude, month and year, field characteristic information like topography and soil quality, and, as you note, a coarse PET layer representing climate. Below, we describe the role of each of these layers and explain why we feel that the use of such a coarse PET product is justified in this setting.

1. Latitude and longitude are important variables in our model, enabling it to discern fine spatial patterns from the densely distributed fallow fields in our dataset. These coordinates inherently capture climate and land characteristics, as reflected in the ET values over fallow lands. Thus, the model’s spatial resolution primarily hinges on these coordinates, similar to how ‘month’ and ‘year’ indicators drive its temporal adaptability. The effectiveness of this approach is evident in Fig. 1, which displays a detailed naturally-occurring ET map without any discontinuities caused by PET or other inputs.
2. We incorporate land characteristics, such as soil quality and topography, to mitigate

potential biases from differences between fallow and agricultural fields. Farmers may preferentially fallow fields with less favorable soil or topography. By explicitly including these variables, the model can adjust for such disparities, ensuring a more accurate and unbiased prediction.

3. Finally, we include PET in our model to enhance its predictive accuracy. While higher resolution PET might slightly improve the model, our current R^2 is 0.87. Considering the minimal variation left unaccounted for and the relative errors in OpenET estimates, further refining PET data resolution is unlikely to significantly impact our agricultural ET estimates.

We have amended our methods section to include a more detailed description of our use of different variables in predicting naturally-occurring ET (line 374):

We predict naturally-occurring ET based on latitude and longitude, the month and year, as well as a broad set of additional variables describing topography (elevation, aspect, slope, topographic wetness index), soil quality (California Storie Index), and climate (Potential ET). The latitude, longitude, and month and year indicators are included to capture the spatial and temporal patterns in ET underlying the densely distributed fallow fields in our dataset. However, the inclusion of an indicator variable for each year in our model limits our analysis to 2016, 2018, and 2019, as these are the only years for which we have available land cover data. Nevertheless, given that 'year' contributes a mere 3% to our model's predictive capability (see Supplementary Fig. 17), it may be considered non-essential for future research in this area.

Including additional predictors in our model presents two benefits. First, they can improve the model's predictive power. For example, we find that Potential ET contributes greatly to the model's final predictions (Supplementary Fig. 17). Other variables are important to include because they can help correct for systematic differences between the fallow lands used to train our model and the active agricultural lands we apply the model to. Such differences could arise from farmers selecting lands to be fallowed due to their inherently lower productivity, which would negatively bias our estimates of counterfactual naturally-occurring ET in locations actively being cropped today. To assemble these predictor variables, we retrieve topographic information from the USGS National Elevations Database, soil quality information from the California Storie Index in the the USDA's gSSURGO and STATSGO2 datasets, and Potential ET from the hPET global dataset (Singer et al., 2021).

2.2 Costs and benefits of different management strategies

2) Concluding that crop switching and/or land fallowing as a scenario for saving water is not novel, PPIC and others have estimated potential water savings. In fact, PPIC extended their analysis to market and other considerations for the growers (<https://www.ppic.org/publication/exploring-the-potential-for-water-limited-agriculture-in-the-san-joaquin-valley/> , <https://www.sciencedirect.com/science/article/pii/S0048969722070632>).

Authors should consider more comprehensive analysis of cost and benefits of different alternatives.

Thank you for your valuable feedback regarding our study’s context within the broader field of research on crop switching and land fallowing for water savings. We highly appreciate your insights and references, which have guided us in refining our manuscript. In particular, your suggestions have led us to more explicitly acknowledge related studies and to expand our discussion to encompass a broader perspective on the costs and benefits of various management strategies. Below, we outline how we’ve integrated these considerations into our manuscript, and also highlight the distinctive contributions of our work.

2.2.1 Our Study’s Unique Contribution

While previous research, including the valuable studies by PPIC and others that you mentioned, has indeed explored water savings through crop switching and fallowing, our work offers a novel approach. Unlike traditional models based on crop coefficients, our data-driven methodology allows for a broader exploration of factors influencing water consumption. This includes variables like orchard age and farming practices, which typically fall outside the scope of conventional models.

Our primary contribution lies in providing a new, data-driven framework to robustly estimate agricultural water consumption. This approach can be used to estimate the water savings potential of different crop switching, fallowing, and farmer’s practices management scenarios, as we demonstrate in our study. It can also be used to conduct future analyses identifying opportunities for water saving across California and beyond. We’ve clarified this in the introduction of our study, setting up our work in direct relation to previous research while emphasizing its innovative aspects (line 58):

Two methodological challenges prevent **agricultural ET from being monitored at scale**. **First, simply measuring total ET at scale is a challenge**. **Second, even when measures of ET are available, it is difficult to separate agricultural ET from it** (Marston et al., 2022; National Academies of Sciences, 2019). **To measure total ET**, eddy covariance flux towers are highly accurate for monitoring ET at a single location (Pastorello et al., 2020). However, they are expensive and thus sparse, and they are designed to measure ET over uniform vegetation, which is not reflective of complex agricultural landscapes (Baldocchi, 2003). In the absence of ET measurements, theoretical water demand can be **simulated** based on climate and crop type (Johansson et al., 2016; Mancosu et al., 2016). Although such tools are helpful for water demand planning (Peterson et al., 2022), theoretical water demand represents the ET of a crop whose water demands are fully met. **Therefore, these are likely to overestimate actual ET**. Additionally, **these simulated estimates exclusively reflect variations in water demand influenced by factors incorporated into the model, which are often limited to crop type and climate** (Orang et al., 2013). Therefore, analyses based on these models are constrained to examining these specific factors, neglecting other critical drivers of ET such as farming practices. Given the scarcity of in situ data and the inherent constraints of simulated estimates, accurately gauging total ET at scale remains a significant challenge.

2.2.2 Enhanced Discussion on Management Strategies

Responding to your suggestion, we've incorporated a discussion on the costs and benefits of different management practices, including insights from the studies you referenced. We've outlined how our findings align with these studies and also where they offer new perspectives, particularly in terms of financial viability and market adaptability of less water-intensive crops in California (line 260):

In situ assessments of crop-specific water consumption and other variations in agricultural ET enable us to model potential water savings across diverse management scenarios. In addition to previously mentioned scenarios based on farming practices, we also investigate more traditionally studied strategies, such as crop switching and fallowing. Consistent with prior studies relying on crop water demand simulations, our research suggests that transitioning to crops with lower water requirements is an effective conservation strategy. However, accruing substantial water savings requires embracing less popular crops like grains and hay (Peterson et al., 2022). The feasibility of increasing production of these crops in the Central Valley is uncertain due to high labor and operational costs (Peterson et al., 2022). Furthermore, any form of crop switching entails expenses related to the adoption of new knowledge, technologies, and market adjustments (Hornbeck, 2012; Kurukulasuriya and Mendelsohn, 2008). Consequently, the viability of a significant shift towards less water-demanding crops, and its alignment with market expansion, remains uncertain (Peterson et al., 2022). Therefore, our findings support the notion that extensive fallowing or land retirement may be essential to achieve substantial reduction targets in areas with severe overdrafts (Hanak et al., 2017; Kelsey et al., 2018). Considering the risk of increased dust from unused land, repurposing such areas for habitat restoration, flood water capture for groundwater replenishment (Guivetchi et al., 2018), solar energy production, or sustainable industrial development (Fernandez-Bou et al., 2023) could mitigate the economic impacts of land retirement for both farmers and local communities. Detailed agricultural ET maps like the ones generated for this study can help determine the scale of land repurposing needed and identify priority areas for such initiatives under different constraints, including existing water rights (Nelson and Burchfield, 2017).

In summary, your feedback has been instrumental in enhancing our manuscript. We believe these revisions provide a clearer understanding of our study's distinct place in ongoing research efforts, as well as a more nuanced discussion of factors influencing water management decisions in California.

2.3 Irrigation efficiency calculation

3) The calculation of irrigation efficiency seems flawed. First, I do not know the details of the USGS delivery data and whether it accounts for water moving in and out of counties.

Looking at the map (Fig. 4), it is hard to comprehend that Mariposa and Madera counties have higher irrigation than Merced and Fresno. San Francisco and Marine counties also jump out. The north south valley split in the irrigation efficiency (Fig. 4, E8) suggests me that USGS data is likely incomplete and this pattern may very well driven by the local and state/federal water deliveries. Or it could be an artifact of your analysis time mismatch. Also, the comparison between theoretical and empirical irrigation efficiency is unfair since theoretical efficiencies do not account for conveyance losses.

Thank you for your critical insights regarding the calculation of irrigation efficiency. We apologize for any confusion caused by the initial presentation of our data and appreciate the opportunity to clarify the USGS data usage in our study.

2.3.1 The nature of the USGS data.

The USGS data estimates irrigation water at the point of use in California, meaning water diverted from a northern county to a southern one is counted in the county where it is used, making it a suitable source for estimating irrigation efficiency. Specifically, from personal correspondence with the USGS: “Regarding water diverted for agriculture in California, the withdrawals for crop irrigation are estimated for the point of use. The irrigation withdrawal points of diversion generally are located in the same county as where the water is applied (used).” We have modified the methods section and the caption to Fig. 4 to better reflect this, and have replaced the word “diversion” with “withdrawal” for improved clarity wherever applicable (e.g. lines 53, 207, 315). For instance, in the methods section, we have (line 479):

To determine **amounts of water withdrawn for irrigation**, we retrieve county level **irrigation water use** data from the USGS National Water Information System. **We note that these irrigation amounts are counted at the point of use, rather than the water’s point of origin.**

2.3.2 A more intuitive map of irrigation.

While the USGS data does account for intra-county diversions, as you note, many of the irrigation numbers displayed in Fig. 4 were counter-intuitive. This is because rather than displaying irrigation by volume, we were displaying irrigation in mm, which is a measure of volume divided by area (area of irrigated land, in our case.) To stave off confusion surrounding this matter, we have altered this figure to display volumetric irrigation as opposed to irrigation depth.

Fig. 4. Irrigation efficiency across the counties of the Central Valley. (a) Irrigation efficiency is calculated by dividing agricultural ET (gridded data) by USGS county-level reports of irrigation amounts (blue polygons). For the calculation, agricultural ET is averaged to the county level to match the spatial scale of the irrigation data. Additionally, irrigation is displayed in volumetric units (teragrams), but is divided by county-level cropland area to be in units consistent with agricultural ET prior to the calculation. (b) The resulting county-level irrigation efficiency estimates vary widely across the Central Valley, with particularly low efficiencies in the northern counties.

This clarification has also been reflected in the methods section (line 488):

ET_{ag} and *Irrigation* must be in matching units, either volumetric or depth. We calculate both in mm per year.

We assume all active agricultural lands in the Central Valley are irrigated and calculate the average agricultural ET in mm per year over active agricultural lands in each county. To also retrieve average irrigation amount across irrigated lands in a county in mm per year, we divide the volume of irrigation water by the average area of irrigated land in each county.

2.3.3 Temporal mismatch effects.

Regarding whether the north-south gradient in irrigation efficiency could be driven by the temporal mismatch in our data, Supplementary Fig. 9 suggests that this is unlikely.

Supplementary Fig. 9. Uncertainty in irrigation efficiency estimates. To calculate irrigation efficiency, we divide agricultural ET by irrigation amount, but the available data do not overlap in years over which they are available. Counties are ordered by latitude, with more northern counties on the left. The average irrigation efficiency is shown as a red x, but we also assess the effect of using different years in the numerator and denominator by combining all different permutations of agricultural ET (shape) and irrigation (color) data. We note that 2016 and 2015 were drought years, while 2018, 2019, and 2010 were wetter. We do find substantial spreads in irrigation efficiency driven by the year of irrigation data, especially in some southern counties. However, most of the spread is driven from combining wet and dry years together, suggesting that an average may successfully decrease much of the variability.

2.3.4 Conveyance losses in theoretical efficiencies.

Finally, we apologize for the confusion on the point about whether theoretical efficiencies account for conveyance losses. We note in the supplementary information, where we carry out this comparison, that we do use a standard 85% conveyance loss rate to calculate theoretical efficiencies (SI line 64).

We set conveyance efficiency to .85, management efficiency to .95, and application efficiency to .60 for flood irrigation, .75 for sprinkler irrigation, and .95 for drip irrigation.

We believe these amendments enhance clarity in our manuscript with respect to the irrigation efficiency calculations. Your expertise in the dynamics of water movement in the Central Valley has been invaluable in refining our approach.

2.4 Improving the discussion section

4) *Discussion section is very superficial and can be improved by referencing prior work on irrigation efficiency paradox, complexities with land fallowing and crop switching, market pressure, water rights, land retirement for other beneficial use like solar, wildlife habitat, and even FloodMAR.*

Thank you for suggesting these pertinent topics, which have greatly strengthened our discussion. As mentioned above, we have amended our discussion section to reference prior work concerning the complexities with land fallowing (line 272-281) and crop switching (lines 265-272), market pressure (lines 270, 272), water rights (line 281), land retirement for other beneficial use like solar (line 276), wildlife habitat (line 275), and FloodMAR (line 276). Additionally, we include a new paragraph detailing the nuances of the irrigation efficiency paradox (lines 282-297). The updated text now reads (line 260):

In situ assessments of crop-specific water consumption and other variations in agricultural ET enable us to model potential water savings across diverse management scenarios. In addition to previously mentioned scenarios based on farming practices, we also investigate more traditionally studied strategies, such as crop switching and fallowing. Consistent with prior studies relying on crop water demand simulations, our research suggests that transitioning to crops with lower water requirements is an effective conservation strategy. However, accruing substantial water savings requires embracing less popular crops like grains and hay (Peterson et al., 2022). The feasibility of increasing production of these crops in the Central Valley is questionable due to high labor and operational costs (Peterson et al., 2022). Furthermore, any form of crop switching entails expenses related to the adoption of new knowledge, technologies, and market adjustments (Hornbeck, 2012; Kurukulasuriya and Mendelsohn, 2008). Consequently, the viability of a significant shift towards less water-demanding crops, and its alignment with market expansion, remains uncertain (Peterson et al., 2022). Therefore, our findings support the notion that extensive fallowing or land retirement may be essential to achieve substantial reduction targets in areas with severe overdrafts (Hanak et al., 2017; Kelsey et al., 2018). Considering the risk of increased dust from unused land, repurposing such areas for habitat restoration, flood water capture for groundwater replenishment (Guivetchi et al., 2018), solar energy production, or sustainable industrial development (Fernandez-Bou et al., 2023) could mitigate the economic impacts of land retirement for both farmers and local communities. Detailed agricultural ET maps like the ones generated for this study can help determine the scale of land repurposing needed and identify priority areas for such initiatives under different constraints such as existing water rights (Nelson and Burchfield, 2017).

Finally, we find a significant opportunity to improve irrigation efficiency, especially in the northern part of the Central Valley where we find lower irrigation efficiency than previously expected (Supplementary Note 2, Supplementary Fig. 7). This suggests that farmers could reduce irrigation water without affecting agricultural ET, and therefore

crop growth. While this aligns with management plans targeting irrigation efficiency improvements (Bruno et al., 2022), improving irrigation efficiency would not necessarily translate to water savings at the watershed scale (Grafton et al., 2018). This is because non-consumed water does not necessarily leave the watershed, and may return to a groundwater reservoir or surface water body and be reused at a later date. On the other hand, the evapotranspired water that makes up the "efficient fraction of irrigation efficiency does leave the watershed entirely (Grafton et al., 2018). Therefore, increasing irrigation efficiency could paradoxically decrease water availability if it is not accompanied by a corresponding decrease in irrigation water withdrawals. Managers could leverage mapped agricultural ET estimates to ensure that water consumption does not increase as irrigation efficiency improvements are rolled out.

2.5 The use of 'empirical' vs. 'modeled'

5) Lastly, I am intrigued by your use of the word *empirical* vs. *modeled*. Example: Agriculture ET derived from CalSIMETAW is being referred as *modeled* but same data from OpenET is *empirical*. Even for the sake of clarity, it would be better to refer the data by source.

Thank you for pointing out our confusing use of the terms 'empirical' and 'modeled'. In order to distinguish between the two kinds of estimates which, as you point out, are both modeled, we ensure CalSIMETAW is referred to by name in both the discussion (lines 221, 226) and supplementary information (SI line 19). We additionally replace the term "modeled" with "simulated" when generally referring to estimates calculated using crop-coefficient based models in order to distinguish from our in situ estimates. We have copied the revised figure from the supplementary information below.

Supplementary Fig. 6. Simulated vs. empirical agricultural ET. Simulated agricultural ET is calculated using theoretical crop water demand from the CalSIMETAW model and by subtracting either (A) naturally-occurring ET estimated using machine learning or (B) precipitation. Each point represents the agricultural ET for a crop in a specific county in the California Central Valley.

2.6 Specific Comments

a) More information should be provided on how various variables are influencing ET_{nat} prediction using ML model. It was briefly mentioned in discussion section (line 240-242) but is not well-defined.

Thank you for your valuable suggestion regarding the need for more detailed information on the influence of various variables in our machine learning model for predicting naturally-occurring ET. Acknowledging the importance of this detail and also in response to your first comment, we have revised the methods section to offer a clearer and more comprehensive explanation of the role each variable plays in the model. We reproduce the corresponding excerpt once more below (line 374):

We predict naturally-occurring ET based on latitude and longitude, the month and year, as well as a broad set of additional variables describing topography (elevation, aspect, slope, topographic wetness index), soil quality (California Storie Index), and climate (Potential ET). The latitude, longitude, and month and year indicators are included to capture the spatial and temporal patterns in ET underlying the densely distributed fallow fields in our dataset. However, the inclusion of an indicator variable for each year in our model limits our analysis to 2016, 2018, and 2019, as these are the only years for which we have available land cover data. Nevertheless, given that 'year' contributes a mere 3% to our model's predictive capability (see Supplementary Fig. 17), it may be considered non-essential for future research in this area.

Including additional predictors in our model presents two benefits. First, they can

improve the model’s predictive power. For example, we find that Potential ET contributes greatly to the model’s final predictions (Supplementary Fig. 17). Other variables are important to include because they can help correct for systematic differences between the fallow lands used to train our model and the active agricultural lands we apply the model to. Such differences could arise from farmers selecting lands to be fallowed due to their inherently lower productivity, which would negatively bias our estimates of counterfactual naturally-occurring ET in locations actively being cropped today. To assemble these predictor variables, we retrieve topographic information from the USGS National Elevations Database, soil quality information from the California Storie Index in the the USDA’s gSSURGO and STATSGO2 datasets, and Potential ET from the hPET global dataset (Singer et al., 2021).

Furthermore, we have included a supplementary figure (Supplementary Fig. 17) to visually illustrate the significance of different variables in the model’s predictions. We hope these enhancements address your concerns and enrich your and future readers’ understanding of our methodology.

Supplementary Fig. 17. The contribution of each variable included in the gradient boosting model on its predictions of naturally-occurring ET.

b) The spatial scale of all analyses differs from each other. ET_{nat} was carried out at 70m resolution, variation in ET at Valley, water saving potential at groundwater sub-basin and irrigation efficiency at county level. More information on how spatial aggregation was done for the ET components from 70m to county/sub-basins scales and reasons for choosing these scales will be helpful.

We apologize for the lack of clarity surrounding the scales of analysis. Below, we explain the choices made surrounding the different scales of analysis and describe changes we have made to the analysis to increase clarity for future readers.

As you mention, we first estimate naturally-occurring ET, and therefore agricultural ET, at sub-field scales to allow us to capture fine scale variation across the Central Valley. We then exploit this sub-field scale dataset to investigate Valley-wide drivers in agricultural ET variation, such as crop type.

We have included a new sentence in the results section explaining this (line 130):

We leverage the significant variability in agricultural ET observed across the Central Valley (Fig. 1) to analyze the factors driving these variations.

Following the calculation of these high level statistics, we conduct different simulations wherein we mimic various management interventions by manipulating sub-field scale agricultural ET to investigate their overall water saving potential. These manipulations are still carried out at the sub-field scale, and therefore necessitate the high-resolution agricultural ET dataset (i.e. we judge whether to simulate a change in agricultural ET based on the properties and agricultural ET of each separate pixel). However, when carrying out these simulations, we calculate crop-specific agricultural ET and within-crop variation at the level of groundwater sub-basins in order to avoid comparing agricultural ET across different climatological contexts. In other words, we avoid prescribing that a farmer in a drier area grow crops with the same agricultural ET as a farmer in a wetter area. The percent water saved by such interventions could be displayed at any number of different spatial scales, but we choose report savings at the groundwater sub-basin level since the interpretation of water savings at the field scale for these scenarios would make less sense from a management perspective, which tends to operate at landscape scales. For example, mapping the water saved by following 5% of plots with the highest level of water consumption would render a map of points that would be difficult to interpret, as opposed to the average water it may save in a particular sub-basin with particular water saving goals.

We edit our results section to better reflect this (line 156):

The variability of agricultural ET both within and between crops allows us to evaluate the water saving-potential of different management strategies. Here, we compare the effect of three scenarios on reducing agricultural ET in groundwater sub-basins across the Central Valley:

1. Crop switching: Substitute high-ET crops for the median water-consuming crop (eq. 6).

2. Farming practices: Keep the same spatial allocation of crops, but reduce agricultural ET of high consumers to the median, crop-specific, consumption level (eq. 7).
3. Fallowing: Fallow the 5% of lands with the highest estimated agricultural ET (eq. 8).

Because it would not make sense to prescribe farmers in different contexts to consume similar amounts of water, we calculate crop-specific agricultural ET and within-crop variation at the level of groundwater sub-basins. We additionally control for differences in climate, topography, and soil type before conducting the scenarios. To account for the effects of orchard age, we remove orchards that have been bearing fruit for 5 years or less or that are in their last year of production (Supplementary Fig. 7).

Finally, we calculate irrigation efficiency at the county level simply because higher-resolution data on irrigation amounts are not available. We simply aggregate our field-scale agricultural ET measurements by taking the mean over the entire county, and divide this by mean irrigation in a given county. We note that since both estimates are in mm per year, they already inherently account for the amount of land irrigated in a given county. As mentioned in our response to your third comment, we have edited the caption to the figure displaying irrigation efficiency to better reflect this and have revisited the methods section for improved clarity on this matter (Fig. 4):

Fig. 4. Irrigation efficiency across the counties of the Central Valley. (a) Irrigation efficiency is calculated by dividing agricultural ET (gridded data) by USGS county-level reports of irrigation amounts (blue polygons). For the calculation, agricultural ET is averaged to the county level to match the spatial scale of the irrigation data. Additionally, irrigation is displayed in volumetric units (teragrams), but is divided by county-level cropland area to be in units consistent with agricultural ET prior to the calculation. (b) The resulting county-level irrigation efficiency estimates vary widely across the Central Valley, with particularly low efficiencies in the northern counties.

We have also clarified this in the methods section (line 487):

ET_{ag} and *Irrigation* must be in matching units, either volumetric or depth. We calculate both in mm per year.

We assume all active agricultural lands in the Central Valley are irrigated and calculate the average agricultural ET in mm per year over active agricultural lands in each county. To also retrieve average irrigation amount across irrigated lands in a county in mm per year, we divide the volume of irrigation water by the average area of irrigated land in each county.

c) Add details on crop groupings shown in Fig. 2 in the supplementary material.

We have included the following passage in the supplementary material to provide a comprehensive explanation of the groupings in Fig. 2 (Supplementary Note 7, SI line 151).

Supplementary Note 7: Crop groups

The crop groups used to make Fig. 2 follow the groupings from the Department of Water Resources LandIQ dataset which we use to designate crop type in our analysis. More information on these groups can be found in the metadata for these data, and are also described here:

1. Grain and hay crops include barley, wheat, oats, miscellaneous grain and hay, and mixed grain and hay.
2. Rice includes rice and wild rice.
3. Field crops include cotton, safflower, flax, hops, sugar beets, corn (field & sweet), grain sorghum, sudan, castor beans, beans (dry), miscellaneous field, sunflowers, hybrid sorghum/sudan, millet, sugar cane, and corn.
4. Pasture includes alfalfa & alfalfa mixtures, clover, mixed pasture, native pasture, induced high water table native pasture, miscellaneous grasses, turf farms, bermuda grass, rye grass, and klein grass.
5. Truck, nursery & berry crops include artichokes, asparagus, beans (green), cole crops, carrots, celery, lettuce (all types), melons, squash, and cucumbers (all types), onions & garlic, peas, potatoes, sweet potatoes, spinach, tomatoes (processing), flowers, nursery & Christmas tree farms, mixed (four or more), miscellaneous truck, bush berries, strawberries, peppers (chili, bell, etc.), broccoli, cabbage, cauliflower, brussels sprouts, tomatoes (market), greenhouse, blueberries, Asian leafy vegetables, lettuce or leafy greens, and potato or sweet potato.
6. Deciduous fruits and nuts include apples, apricots, cherries, peaches and nectarines, pears, plums, prunes, figs, miscellaneous deciduous, mixed deciduous, almonds, walnuts, pistachios, pomegranates, and plums, prunes or apricots.
7. Citrus and subtropical include grapefruit, lemons, oranges, dates, avocados, olives, miscellaneous subtropical fruit, kiwis, jojoba, eucalyptus, and mixed subtropical fruits.
8. Vineyards include table grapes, wine grapes, and raisin grapes.

d) Fig. A4 can be represented better with contrasting colors, hard to see the overlap. Also, plots with combination of bar and box plots can be improved with different color choices and adding gridlines.

We have implemented your suggestions surrounding the readability and accessibility of our figures in full.

First, we have increased the brightness of the colors in the bar plot/box plot figures, and added gridlines.

Fig. 2. Variations in annual agricultural ET across and within crop groups. Mean agricultural ET by crop group (green fill and 95% CI) is the average difference between total ET (black outline) and naturally-occurring ET (brown fill). All measures are summed across the year, leading to naturally-occurring and total ET estimates that include water consumption occurring outside of the growing season (Supplementary Figure 6). While we find significant differences in mean agricultural ET across crop groups, the gray box plots also show a broad spread in agricultural ET within crop groups (box plots show 0.5, 0.25, 0.5, 0.75, and 0.95 quantiles).

Additionally, we have changed the colors in Figure A4 to enhance contrast. However, we note that Figure A4 shows lines that are closely overlapping, highlighting the close matchup between our machine learning model predictions and OpenET estimates of naturally-occurring ET over fallow lands but also making them inherently difficult to distinguish.

Supplementary Fig. 4. Temporal error structure of ET predictions over fallow lands. The average ET observed and predicted over fallow lands are similar throughout the year. In other words, the model is not more likely to be positively or negatively biased based on the month it is predicting over.

e) The title seems misleading as the field-scale estimates are based of remotely sensed and other data of different scales, aggregated for the suggested water saving strategies at county/groundwater sub-basin scales. You may want to consider revising it.

Thank you for your insights surrounding the accuracy and descriptiveness of our title. After much deliberation, we have elected to keep the same title as we feel that it captures the two main contributions of our work: that we are able to estimate agricultural ET at the field scale and that these fine resolution estimates allow for various analyses that may be used for water management.

As we mention above in addressing your point on the different scales of our analyses, our crop switching, fallowing, and farmer practices management scenarios do not aggregate the field-scale data to the groundwater sub-basin scale. On the contrary, all of the fine spatial variation we uncover when calculating our high-resolution agricultural ET estimates is necessary and utilized when performing these analyses, as can be seen in equations 5-7. The results are reported at a coarser spatial scale (the groundwater sub-basin) for ease of interpretation.

However, we do average our field-scale estimates of crop water consumption to the county level when calculating irrigation efficiency since higher resolution information on water accessed for irrigation is not available. This, however, does not change the potential of our

field scale estimates to be used to do finer scale analyses on irrigation efficiency in the future as higher resolution water use data becomes available.

We therefore choose to retain our title, “Field-scale crop water consumption estimates reveal potential water savings in California agriculture,” but thank you for the opportunity to revisit it and consider its clarity and accuracy.

f) Some sentences are lengthy and can be further simplified in the Introduction section. For example: lines 44-47, 68-71, 51-53.

We have carefully read through and revised the lines you mentioned as well as several others. Some examples are included below (lines 46, 52, 58, 64, 75, 79).

Agricultural ET, or the increase in ET that irrigated agriculture brings, is **therefore** a critical measure in that it represents the amount of water that is actually “consumed” by agriculture. **This water leaves the watershed entirely as it is evaporated from the soil and transpired by crops** (Hoekstra and Mekonnen, 2012).

When compared to total irrigation amounts, agricultural ET can also be used to highlight the fraction of irrigation withdrawals that do not effectively result in consumptive use (Puy et al., 2022). **While this water stays in the system and therefore may continue to provide beneficial uses** (Grafton et al., 2018), it also does not achieve its original purpose of contributing to crop growth. **Therefore, agricultural ET can help calculate irrigation efficiency and identify unnecessary irrigation water withdrawals from surface or groundwater reservoirs** (Puy et al., 2022).

Two methodological challenges prevent **agricultural ET from being monitored at scale**. **First, simply measuring total ET at scale is a challenge**. **Second, even when measures of ET are available, it is difficult to separate agricultural ET from it** (Marston et al., 2022; National Academies of Sciences, 2019).

In the absence of ET measurements, theoretical water demand can be **simulated** based on climate and crop type (Johansson et al., 2016; Mancosu et al., 2016). Although such tools are helpful for water demand planning (Peterson et al., 2022), theoretical water demand represents the ET of a crop whose water demands are fully met. **Therefore, these are likely to overestimate actual ET**.

Addressing the second challenge, even when accurate ET measurements are available, isolating agricultural ET from total ET is **difficult**. **This step is key for water management, since the resulting metric represents the increase in ET attributable to agriculture**.

Most often, this naturally-occurring ET is assumed to be equal to precipitation (Mancosu et al., 2016; Snyder et al., 2012). **However**, there may be temporal lags between precipitation and resulting ET, some precipitation may not result in ET at all, and ET

may also be supplemented by other sources of water, such as near surface groundwater.

g) Given how the ML model was developed, should you be still calling the ET over fallow land as natural? Don't you think fallow ET would be more appropriate?

Thank you for your observation regarding the terminology we used to denominate ET over fallow land. Choosing between ‘fallow ET’ and ‘naturally-occurring ET’ was a nuanced decision in our study, carefully considered to best reflect our research framework. We ultimately opted for ‘naturally-occurring ET’ as it aligns more coherently with the overarching conceptual approach of our work, rather than emphasizing a specific methodological aspect.

In line with this overarching framework, we write in the methods section (line 346):

We note that another, theoretically distinct counterfactual could be constructed to represent ET_{nat} : ET if the land were undisturbed, natural land rather than fallow. We elect to simulate ET over fallow lands since it allows us to predict the potential water savings from fallowing, **which has a management relevant interpretation.**

2.7 Line-specific comments

L46: evapotranspiration is “both” evaporation and transpiration that often occur simultaneously.

We have edited to more clearly portray evapotranspiration (line 46):

Agricultural ET, or the increase in ET that irrigated agriculture brings, is **therefore** a critical measure in that it represents the amount of water that is actually “consumed” by agriculture. **This water leaves the watershed entirely as it is evaporated from the soil and transpired by crops (Hoekstra and Mekonnen, 2012).**

L111: “R2”

Thank you, this is fixed (line 124).

L124: More information on how 34% was calculated will be helpful for readers.

Thank you for pointing out the need for clarity on how we calculated the 34% figure. We have added detailed information about this calculation in the methods section to enhance the manuscript’s transparency and aid reader understanding (line 443):

In addition to using regression to calculate point estimates and confidence intervals, regression allows us to calculate the proportion of variation that is explained by a set of variables. This is because the R^2 corresponds to the fraction of variation explained by the regression. We use this to assess the % variation explained by crop type, using

eq. 4 [...]

L125: “water intensive crops”

We have amended the language of this line for improved clarity (line 134):

In Fig. 2, we show the water intensity of different crop groups (Supplementary Note 7).

L128: reference to “(Figure A1)” seems wrong.

Thank you, this is fixed (line 137).

L207-208: You can write that “. . . 30% deficit irrigation for some crops (wheat, maize, cotton etc.)” as the referred studies are only for limited crops.

We have removed this statement from the paper entirely, and replaced it with a more nuanced discussion of different kinds of farmer practices (line 236):

High-resolution, empirical estimates of agricultural ET additionally open up the possibility to investigate differences in agricultural ET past what is attributable to crop type, which we find only accounts for 34% of the variation. These within-crop variations can be substantial for many crops: the difference in agricultural ET of a pistachio field from the 75th to the 25th percentile is the same as the water that could be saved from fallowing an alfalfa field. Such broad variability is consistent with findings from studies comparing total ET across crops during the growing season Schauer and Senay (2019); Wong et al. (2021). Even after adjusting for variability attributable to field characteristics, orchard age, and climate, we find that these variations could translate to substantial water savings without requiring a change in crop type. Unfortunately, a lack of high-resolution data on field-scale farming practices and yields inhibits further analysis of specific practices driving agricultural ET variation and their economic implications. However, prior literature from experimental plots or particular locations suggests that mulching (Kader et al., 2019), conservation tillage (Mitchell et al., 2012), deficit irrigation (English and Raja, 1996; Goldhamer, 1999; Rudnick et al., 2019), and improved irrigation scheduling and technologies (Jovanovic et al., 2020) all have potential to limit agricultural ET. These practices may prove advantageous relative to costly strategies like crop-switching or fallowing, though more detailed cost-benefit analyses are necessary to determine the suitability of various interventions in specific contexts (Mitchell et al., 2016). Such water saving farming practices are not mentioned in the plans drafted by SGMA water managers (Bruno et al., 2022). This is possibly due to a lack of conclusive research on the potential of such strategies to effectively decrease agricultural ET without significant effects on yield or operation cost. Spatial data on the use of these different practices would allow researchers to take full advantage of our high-resolution agricultural ET estimates and study their water-saving and economic

benefits.

L211-214: The point made here makes sense, but not sure how relevant is the precipitation factor for Central valley agriculture as the peak demand and rainfall occurs are different times of the year.

We appreciate your observation regarding the relevance of the precipitation counterfactual given the mismatch in seasonality with the crop growing season. It appears our initial explanation did not clearly convey our methodology. We actually accounted for precipitation at the monthly level and only included amounts not exceeding crop water demand. To clarify this, we've amended our explanation for why the increased bias occurs (line 226).

We find that the bias between our estimates and CalSIMETAW estimates increases when precipitation, rather than naturally-occurring ET, is used to represent baseline amounts of available water. This can be explained by the seasonal mismatch between precipitation and naturally-occurring ET: although annually there is more precipitation than naturally-occurring ET, precipitation mainly occurs in the winter when it is unavailable to most crops. Because using precipitation as a proxy for naturally-occurring ET does not account for moisture that remains in the soil by the time the growing season begins, this inflates simulated estimates of crop water demand.

Appendix C: Provide citation for CalSIMETAW

Thank you, this is fixed (line 21).

3 Reviewer #3

Review of manuscript titled: “Field-scale crop water consumption estimates reveal potential water savings in California agriculture”

Summary: Authors present a study where through satellite remote sensing and machine learning they can differentiate between naturally-occurring evapotranspiration (ET) and agricultural ET (the increase in ET induced by irrigation) to better quantify agricultural water use in the state of California; and by doing so offer potential solutions to decrease agricultural water use. This study is timely and relevant and takes advantage of the emerging technologies and approaches now available to earth system scientists. The scope of the manuscript is ambitious, but the authors have done a good job of explaining the steps and approaches/assumptions taken. I do believe the approach taken is novel (the subtraction of the naturally-occurring ET) but am not entirely convinced of its validity. Other studies have quantified crop water use over the state using ET (Wong et al., 2021; Schauer et al., 2019; among others), but are not discussed in the manuscript. This may simply be due to the writing constraints of Nature Communications, but I still believe it is important to bring up.

Overall, I think this is a wonderful and important study. But I do have some general concerns that I think should be addressed prior to publication.

3.1 Estimating agricultural ET

Concern #1. This is a novel approach. Quantifying agricultural ET by determining the ET that would be naturally-occurring in that location should the crop (and applied irrigation) not be present and subtracting it from the actual ET observed via OpenET. That said, I’m hesitant to agree that the definition of agricultural ET is this difference. My hesitation stems from the results stating that, on average/in general, 46% of total ET is defined as naturally-occurring and 53% of total ET is defined as agricultural ET. Intuitively this seems very high for ‘naturally-occurring’. If I were in an almond orchard in September, I would not expect that 46% of the total ET coming off the field would be naturally-occurring and only 53% would be because of the almond tree and the applied irrigation. I know you have broken these percentages down into crop categories, but they still seem high. I’m also aware that there are likely some time dependency factors that play a role. That said, I’m curious if you have any citations confirming this is a viable assumption or if you have ground data that can be used to validate your assumption. Also, because the idea behind the study is to quantify agricultural ET, should there be more of a month-by-month analysis or growing season analysis? I think it is important prior to scaling an approach like this to the entire state and making claims about water use that a proof of concept be completed, either on the ground or found in the literature.

We appreciate your comment which has prompted us to think more deeply about our definition of agricultural ET and to inquire into the accuracy of our estimates of naturally-occurring ET. Below, we break our answer into several sections. First, we consider our definition of agricultural ET and its appropriateness in the context of our study. We compare it to the methodology employed by Wong et al. (2021) and Schauer and Senay (2019)

that you mention above. We argue that our definition is relevant for water management applications. Second, through a month-by-month analysis, we demonstrate how this definition allows us to calculate annual measures of agricultural ET without relying on external information about the growing season. Third, in response to your comment concerning the seemingly high estimates of naturally-occurring ET, we recognize that reporting annual values of naturally-occurring ET can be confusing since readers may be expecting values confined to the growing season. We therefore adjust our language to improve clarity surrounding this point. We additionally investigate possible sources of error in our estimates of naturally-occurring ET to ensure their accuracy, but do not find evidence of bias.

3.1.1 Our definition of agricultural ET

We define agricultural ET as the increase in ET that agriculture brings. This definition recognizes that not all ET over agricultural lands, during the growing season or otherwise, can necessarily be attributed to agriculture. This definition is particularly useful from a management perspective, since it denotes the decrease in ET, or water savings, that one might expect if agriculture were to cease. Defining agricultural ET in this way implies a counterfactual ET that would be present regardless of agriculture, which we call naturally-occurring ET.

Our conceptualization of agricultural ET differs from the approach of attributing all ET during the growing season over an agricultural field solely to agriculture (Schauer and Senay, 2019; Wong et al., 2021). We believe our definition offers practical advantages for management, especially in terms of identifying potential water savings if agriculture is ceased. Moreover, it addresses the transitional phase between growing seasons, avoiding the ambiguity of classifying ET as agricultural or not in early growing seasons when crops are young and unlikely to be the sole drivers of ET. By considering the incremental impact of agriculture on ET throughout the growing season, our definition remains applicable throughout the entire year, a topic we will explore in the following section.

We edit our manuscript to further clarify our definition of agricultural ET. In the introduction, we have (line 75):

[...] even when accurate ET measurements are available, isolating agricultural ET from total ET is difficult. This step is key for water management, since the resulting metric represents the increase in ET attributable to agriculture. To do so, one must estimate the naturally-occurring ET that would occur in the absence of irrigated agriculture (for example, if the land were left fallow).

Additionally, in the methods section, we note (line 338):

We define agricultural ET as the difference between total ET over an agricultural parcel and the ET that would have been, had that parcel been fallow land instead (eq. 1). This definition recognizes that not all ET over agricultural lands, during the growing season or otherwise, can necessarily be attributed to agriculture. Such a definition is

particularly useful from a management perspective, since it denotes the decrease in ET, or water savings, that one might expect if agriculture were to cease.

$$ET_{ag} = ET_{tot} - ET_{nat} \quad (1)$$

where ET_{ag} is agricultural ET, ET_{tot} is the total ET over an agricultural parcel, and ET_{nat} is the counterfactual ET that would occur naturally, were the same land fallow.

We also edit the manuscript to more effectively put our study in the context of previous work such as that of Wong et al. (2021) and Schauer and Senay (2019). In the introduction, we write (line 87):

Recent advances in the remote sensing of ET unlock new avenues for research into agricultural water consumption. Numerous algorithms for estimating ET using land surface temperature and other remote sensing inputs have been validated specifically for use in agricultural settings (Anderson et al., 2021; Melton et al., 2022). **These advancements have allowed researchers to empirically study total ET in agricultural settings (Schauer and Senay, 2019; Wong et al., 2021).** However, despite high-resolution maps of ET in agricultural landscapes (Anderson et al., 2018; Wong et al., 2021), the challenge of isolating agricultural ET from total ET has been limited to studies in extremely arid regions where irrigation is the only source of water available to plants (Al-Gaadi et al., 2022).

Additionally, in the discussion, we write (line 236):

High-resolution, empirical estimates of agricultural ET additionally open up the possibility to investigate differences in agricultural ET past what is attributable to crop type, which we find only accounts for 34% of the variation. These within-crop variations can be substantial for many crops: the difference in agricultural ET of a pistachio field from the 75th to the 25th percentile is the same as the water that could be saved from fallowing an alfalfa field. Such broad variability is consistent with findings from studies comparing total ET across crops during the growing season (Schauer and Senay, 2019; Wong et al., 2021).

3.1.2 Annual estimates of agricultural ET

In addition to its management relevance, another benefit of our definition of agricultural ET is that summing over the entire year retrieves annual agricultural ET without requiring any assumptions about when the growing season starts or ends. Fig. R1 shows the seasonality of agricultural ET. This figure demonstrates how in the winter, outside of the growing season of the vast majority of crops in the Central Valley, naturally-occurring ET and total ET are nearly identical, leading to a near-zero agricultural ET. As a result, we use annual estimates of agricultural ET in our analyses.

Fig. R1. Time series of agricultural ET in the California Central Valley. Total ET (black) can be partitioned into agricultural ET (green), and naturally-occurring ET (brown). Agricultural ET peaks in the summer, which constitutes the growing season for most crops grown in the Central Valley, but diminishes to nearly zero in the winter when few crops are grown.

3.1.3 Improving clarity surrounding the annual presentation of naturally-occurring ET

We recognize that our estimates of naturally-occurring ET can seem high because they are aggregated over the entire year and include naturally-occurring ET that occurs at times where the water is unavailable to crops. In order to stave off confusion and because they have little management relevance, we have removed statements comparing agricultural ET and naturally-occurring ET amounts. We additionally edit the caption of Figures 1 and 2 to clarify that not all naturally-occurring ET necessarily translates to crop growth. For example, we have copied Fig. 2 below.

Fig. 2. Variations in annual agricultural ET across and within crop groups. Mean agricultural ET by crop group (green fill and 95% CI) is the average difference between total ET (black outline) and naturally-occurring ET (brown fill). All measures are summed across the year, leading to naturally-occurring and total ET estimates that include water consumption occurring outside of the growing season. While we find significant differences in mean agricultural ET across crop groups, the gray box plots also show a broad spread in agricultural ET within crop groups (box plots show 0.5, 0.25, 0.5, 0.75, and 0.95 quantiles).

3.1.4 Sources of uncertainty in estimating agricultural ET

Further responding to your concern that our estimates of naturally-occurring ET may be inflated, we investigate multiple potential sources of error. To ensure the accuracy of our findings and improve the clarity of our work, we follow the following steps, which are described in detail below:

1. **Investigation of OpenET and ML model errors.** We investigate potential errors associated with OpenET and our machine learning model, and find no reason to suspect biased estimates arising from either. We undertake an analysis fully detailing the role of both in our estimates of agricultural ET (Supplementary Note 4).
2. **Analysis of land cover data accuracy.** We confirm that our estimates of naturally-occurring ET are not positively biased due to inaccuracies in our land cover data, causing us to train our model not only on fallow lands, but some active agricultural lands (Supplementary Note 6).
3. **Comparison with other estimates.** We find our estimates of naturally-occurring ET compare well with our estimates of precipitation and estimates of natural ET in Mediterranean climates.

4. **Robustness of findings against hidden biases.** We consider how our analyses would be affected if another source of bias not accounted for in the above analyses were affecting our estimates of naturally-occurring ET. Because our main findings are relative, they would remain unaffected.

Investigation of OpenET and ML model errors. OpenET estimates have been validated over a broad variety of land cover types. According to OpenET’s Intercomparison and Accuracy Assessment Report, croplands, grasslands, and shrublands, all which may help represent the kinds of errors we might expect over fallow lands, exhibit no statistically significant biases, and therefore do not suggest that OpenET is positively biased over fallow lands (Table R 1) (OpenET, 2021). Shrublands may especially be considered as a good proxy for the fallow lands of the Central Valley since the average ET of the shrublands used in the OpenET validation, 31.07 mm/month, is similar to what OpenET predicts over fallow lands in the Central Valley (mean = 36.5 mm/month).

Land cover type	Statistic	Ensemble	Range
Croplands 45 sites, N = 1682 months Mean station ET = 93.68 (mm)	Slope	0.95	0.86 - 1.04
	MBE (mm)	-3.64 (-3.9%)	-13.77 - 5.16 (-14.7% - 5.5%)
	MAE (mm)	155.66 (16.6%)	17.96 - 22.92 (19.2% - 24.5%)
	RMSE (mm)	195.17	23.43 - 25.89 (25% - 27.7%)
	R-squared	0.95	0.89 - 0.93
Grasslands 20 sites, N = 672 months Mean station ET = 42.56 (mm)	Slope	1.03	0.73 - 1.28
	MBE (mm)	-1.23 (-2.9%)	-11.88 - 9.73 (-27.9% - 22.9%)
	MAE (mm)	19.45 (15.4%)	20.17 - 28.1 (47.4% - 66.0%)
	RMSE (mm)	24.12 (56.7%)	24.62 - 35.96 (57.8% - 84.5%)
	R-squared	0.75	0.54 - 0.8
Shrublands 24 sites, N = 681 months Mean station ET = 31.07 (mm)	Slope	0.95	0.65 - 1.34
	MBE (mm)	2.89 (9.3%)	-5.2 - 18.7 (-16.7% - 41.4%)
	MAE (mm)	15.68 (50.5%)	17.45 - 22.64 (56.2% - 72.9%)
	RMSE (mm)	19.96 (64.2%)	20.9 - 29.18 (67.3% - 93.9%)
	R-squared	0.66	0.3 - 0.69

Table R 1. OpenET monthly performance over croplands, grasslands, and shrublands. MBE denotes mean bias error, and none of the land cover types show statistically significant MBE. Reproduced from the OpenET Intercomparison and Accuracy Assessment Report, Table 3 (pg.6).

To improve transparency surrounding the effect of errors inherent to our method, we have added a section in the supplementary information detailing the role of error in our analysis (Supplementary Note 4, SI line 76).

Supplementary Note 4: The effect of OpenET and machine learning model error on agricultural ET estimates

We define agricultural ET as the difference between total ET and naturally-occurring ET, $ET_{ag} = ET_{tot} - ET_{nat}$ (Main text eq. 1), where naturally-occurring ET is the counterfactual ET that would occur naturally, were the same land fallow. However, we calculate agricultural ET using estimates of total ET from OpenET and modeled naturally-occurring ET trained on OpenET data (Supplementary eq. 1). Since the OpenET data and the model we use to estimate naturally-occurring ET have errors associated with them, our estimates of agricultural ET can be described as the true agricultural ET plus several error terms (Supplementary eq. 4), which we derive as follows:

$$ET_{ag}^* = E\hat{T}_{tot} - E\tilde{T}_{nat} \quad (1)$$

where ET_{ag}^* denotes our estimate of agricultural ET, $E\hat{T}_{tot}$ is OpenET's estimate of ET_{tot} and $E\tilde{T}_{nat}$ is our estimate of the naturally-occurring ET, predicted by our machine learning (ML) model. The hat notation indicates estimates inclusive of OpenET error, while the tilde indicates estimates inclusive of ML model error.

Because our naturally-occurring ET ML model is trained on and therefore predicts OpenET values over fallow lands, we can rewrite the same expression as:

$$ET_{ag}^* = E\hat{T}_{tot} - (E\hat{T}_{nat} + \epsilon_{ML}) \quad (2)$$

where $E\hat{T}_{nat}$ is the (naturally-occurring) ET that OpenET would observe if the field were fallow, and ϵ_{ML} is the error from the ML model in predicting $E\hat{T}_{nat}$.

We can then again reorganize the same equation by separating the error from the OpenET observations. This reveals the relationship between the true ET_{ag} and our estimate, ET_{ag}^* :

$$ET_{ag}^* = ET_{tot} + \epsilon_{OpenET,ag} - (ET_{nat} + \epsilon_{OpenET,fal} + \epsilon_{ML}) \quad (3)$$

$$= ET_{ag} + \epsilon_{OpenET,ag} - \epsilon_{OpenET,fal} - \epsilon_{ML} \quad (4)$$

where ET_{ag} , ET_{tot} , and ET_{nat} are the true (unobservable) agricultural ET, total ET, and naturally-occurring ET, respectively, $\epsilon_{OpenET,ag}$ is the error associated with the OpenET estimate over the agricultural field and $\epsilon_{OpenET,fal}$ is the error associated with what the OpenET estimate would be if the land were fallow.

In our study, we seek to estimate mean ET_{ag} , be it over the entire Central Valley, for a certain crop type, or for a given county, we seek to calculate the expectation of ET_{ag} over a sample of grid cells. Using Supplementary eq. 5, we can show that our estimate of $\mathbb{E}[ET_{ag}^*]$ depends not only on $\mathbb{E}[ET_{ag}]$, but also the expectation of the error terms.

$$\mathbb{E}[ET_{ag}^*] = \mathbb{E}[ET_{ag}] + \mathbb{E}[\epsilon_{OpenET,ag}] - \mathbb{E}[\epsilon_{OpenET,fal}] - \mathbb{E}[\epsilon_{ML}] \quad (5)$$

We show in Supplementary Figures 1-4 that ϵ_{ML} is unbiased, that is, $\mathbb{E}[\epsilon_{ML}] = 0$. OpenET has also been shown to produce unbiased estimates over agricultural lands, so $\mathbb{E}[\epsilon_{OpenET,ag}] = 0$ (see Table 3 on page 6 of the OpenET Intercomparison and Accuracy report) (OpenET, 2021). We can also assume $\epsilon_{OpenET,fal}$ is unbiased because though OpenET has not specifically been evaluated over fallow fields, it produces unbiased estimates over natural shrublands and grasslands with ET values of similar magnitude to fallow fields in the Central Valley (OpenET, 2021). As a result, in expectation our estimates of agricultural ET are unbiased, and consequently produce unbiased regression coefficients.

$$\mathbb{E}[ET_{ag}^*] = \mathbb{E}[ET_{ag}] \quad (6)$$

While our estimates of agricultural ET are unbiased in expectation, the error terms do add variance that is not present in the true agricultural ET. Therefore, any analysis that assesses variation across pixels will reflect inflated variance relative to true agricultural ET. For example, this has implications for the farming practices and following scenarios we conduct (Fig. 3). However, here we estimate that error is responsible for only 11% of the variance in our annual estimates of agricultural ET, suggesting its influence over key results is limited. Specifically, if we assume that all error terms are independent, we have:

$$var(ET_{ag}^*) = var(ET_{ag}) + var(\epsilon_{OpenET,ag}) + var(\epsilon_{OpenET,fal}) + var(\epsilon_{ML}) \quad (7)$$

where $var(\cdot)$ denotes the variance.

Then, because variance is equal to mean squared error (MSE) when bias is 0, we have:

$$var(ET_{ag}^*) = var(ET_{ag}) + MSE(\epsilon_{OpenET,ag}) + MSE(\epsilon_{OpenET,fal}) + MSE(\epsilon_{ML}) \quad (8)$$

We calculate the MSE of our machine learning model using our test set, and OpenET provides root MSE values over croplands and shrublands (which we use to approximate error over fallow lands) (again, see Table 3 on page 6 of the OpenET Intercomparison and Accuracy report) (OpenET, 2021). We find that for estimates of yearly ET_{ag}^* , $MSE(\epsilon_{OpenET,ag}) + MSE(\epsilon_{OpenET,fal}) + MSE(\epsilon_{ML})$ represents only 11% of $var(ET_{ag}^*)$. This indicates that the vast majority of the variance we uncover is representative of true differences in water consumption across fields, as opposed to variability due to model errors.

Analysis of land cover data accuracy. The only remaining source of error which could bias our estimates of agricultural ET would be if the land cover data we use to identify fallow lands were imprecise. If this were the case, we would likely be training our naturally-occurring ET machine learning model on agricultural fields. This would lead to an overestimation of naturally-occurring ET and therefore an underestimation of agricultural ET.

To rule out this potential issue, we test the effect of train a model exclusively on land that was marked fallow not only by the DWR dataset we use for our main results, but also by the Cropland Data Layer (CDL). We find that the results do not change significantly with this more stringent criterion, and report these findings in our supplementary information (Supplementary Note 6, Supplementary Fig. 13, 14, and 15) (SI line 142).

Supplementary Note 6: Results when using training data also marked fallow by the Cropland Data Layer (CDL)

When training the naturally-occurring ET model, we rely on the California Department of Water Resources (DWR) LandIQ crop type dataset due to its high accuracy. Here, we evaluate the effect of instead training the data only on fallow fields that are marked as fallow by both the DWR dataset and the Cropland Data Layer (CDL). Overall, this change does not affect our estimates of naturally-occurring ET: while we estimate 438 mm per year (394.4-481.7 95% CI) in the main text, we estimate 423.9 mm per year (369-478.7 95% CI) here. All general trends and conclusions are robust to this change (Supplementary Fig. 14, Fig. 15, Fig. 16).

Supplementary Fig. 14. Variations in agricultural ET across and within crop groups using training data also marked fallow by the Cropland Data Layer. This is the equivalent of Fig. 2 in the main text, but calculated using a naturally-occurring ET model trained on data also marked fallow by the Cropland Data Layer.

Supplementary Fig. 15. The percent reduction in agricultural ET driven by various management scenarios using training data also marked fallow by the Cropland Data Layer. This is the equivalent of Fig. 3 in the main text, but calculated using a naturally-occurring ET model trained on data also marked fallow by the Cropland Data Layer.

Supplementary Fig. 16. Irrigation efficiency across the counties of the Central Valley using training data also marked fallow by the Cropland Data Layer. This is the equivalent of Fig. 4 in the main text, but calculated using a naturally-occurring ET model trained on data also marked fallow by the Cropland Data Layer.

Comparison with other estimates. We additionally note that the numbers we uncover for naturally-occurring ET are within reason when compared to precipitation within the Central Valley: while CalSIMETA reports an average precipitation of 554 mm/year, our naturally-occurring ET estimates amount to 438 mm/year (394.4-481.7 95% CI) (Orang et al., 2013). Additionally, we find that 438 mm/year falls within the range of values found in the literature studying natural systems in Mediterranean climates, which typically report values between 300 and 480 mm/year (Aires et al., 2008; Baldocchi et al., 2004; Duell, 1988).

Robustness of findings against hidden biases. These additional analyses do not suggest that OpenET or other potential sources of error would be inflating our estimates of naturally-occurring ET. However, out of an abundance of caution, we consider how our findings would be affected if our estimates of naturally-occurring ET were nevertheless inflated, causing artificially low estimates of agricultural ET. Because our main findings focus more on relative differences in agricultural ET, the patterns we uncover would remain unchanged. For example, the ranking in water intensity of crops would not be affected by a uniform underestimation of agricultural ET. Our scenarios all report water savings in percent, and would also remain unchanged. Our estimates of irrigation efficiency would increase, but our main finding surrounding the pronounced north to south gradient would not be affected.

3.2 Irrigation diversions

Concern #2. I apologize if I missed this in the manuscript, or if it is common knowledge, but could you please provide a thorough description of “diversion” that is used throughout the manuscript in reference to water diversions and irrigation. My interpretation is that diversion is water found/stored in one location (i.e., county) and “diverted” to another location (i.e., another county). This interpretation makes sense when looking at Fig. 4 (left column) where we find higher water diversions for irrigation in the north (the Sacramento River Valley and north). Water in the northern portion of the state is famously (or infamously, depending on where in the state you are) moved further south for agricultural irrigation. In my mind this makes the southern counties much less efficient (in terms of irrigation efficiency) compared to the north. However, the opposite is presented, with the southern counties reporting higher irrigation efficiencies. Could you please offer more context to this Figure and the overall implications made from the associated analysis?

We apologize for the lack of clarity surrounding the nature of the irrigation data we are using and our use of the word ‘diversion.’ The USGS data in California we use estimates irrigation water at the point of use, so water diverted from a northern county for use in a southern county would be counted in the southern county. Specifically, from personal correspondence with the USGS: “Regarding water diverted for agriculture in California, the withdrawals for crop irrigation are estimated for the point of use. The irrigation withdrawal points of diversion generally are located in the same county as where the water is applied (used).” We have modified the methods section and the caption of Fig. 4 to better reflect this, and have replaced the word “diversion” with “withdrawal” for improved clarity wherever applicable (e.g. lines 53, 207, 315). For example, in the methods section, we have (line 479):

To determine **amounts of water withdrawn for irrigation**, we retrieve county level **irrigation water use** data from the USGS National Water Information System. **We note that these irrigation amounts are counted at the point of use, rather than the water's point of origin.**

However, as you note, many of the numbers displayed in Fig. 4 are counter-intuitive. This is because rather than displaying irrigation by volume, we were displaying irrigation in mm, which is a measure of volume divided by area (area of irrigated land, in our case.) To stave off confusion surrounding this matter, we have altered this figure to display volumetric irrigation as opposed to irrigation depth.

Fig. 4. Irrigation efficiency across the counties of the Central Valley. (a) Irrigation efficiency is calculated by dividing agricultural ET (gridded data) by USGS county-level reports of irrigation amounts (blue polygons). For the calculation, agricultural ET is averaged to the county level to match the spatial scale of the irrigation data. Additionally, irrigation is displayed in volumetric units (teragrams), but is divided by county-level cropland area to be in units consistent with agricultural ET prior to the calculation. (b) The resulting county-level irrigation efficiency estimates vary widely across the Central Valley, with particularly low efficiencies in the northern counties.

3.3 Orchard age and limitations of the farming practices scenario

Concern #3. Different parts of the manuscript discuss the within crop variability in ET as well as statements alluding to how agricultural water applications can be significantly decreased (from 95 percentile to 25 percentile) without changing the landcover. First, a likely reason for ET variability within the same cropping system (particularly tree nuts) is because of the age of the orchard. These systems are perennials and exist for 15-25 years. It makes sense that an annual would present lower variation – it’s effectively the same crop every year. A pistachio orchard changes drastically from 2nd leaf to 20th leaf, with a considerable amount of increase to biomass and by proxy ET. In order to make the claim that water savings can occur without changing the crop, there needs to be an accounting of orchard age. Second, I do not think that scenario #2 in reducing agricultural ET is a viable scenario. Sure, deficit irrigation is possible and is well studied in certain cropping systems. In some, such as wine grapes, its required to produce a high-quality grape. But to suggest cutting back irrigation based on ‘high’ ET estimates to ‘median’ ET estimates without including age (again, perennials introduce some difficulty here), production goals of the commodity, or management style seems cavalier.

This comment raises important points surrounding the depth and accuracy of our analysis, particularly in relation to orchard age and the viability of proposed water-saving scenarios. In response to your concerns, we have investigated the impact of orchard age on agricultural ET and adjusted our analyses to account for this factor. Additionally, we have modified our language to reflect the inherent limitations of the farming practices scenario related to its potential costs, benefits, and uncertainties. We detail these two changes below.

3.3.1 Orchard age

We would first like to thank you for raising this important point about the effect of orchard age on agricultural ET. After investigating this effect in our data and find a large and clear effect of orchard age on agricultural ET, and document this in a new section in our supplementary information (Supplementary Note 2, line 42).

Supplementary Note 2: The effect of orchard age on agricultural ET

Water consumption is known to be affected by orchard age, especially for young orchards (Drechsler et al., 2022). One advantage of data-driven, field-scale estimates of agricultural ET is that they can capture such variation.

Since information about orchard age is not currently available in California, we use the LandIQ dataset’s transitions from “young perennial” classifications to crop-specific orchard categories as a proxy for orchard age. “Young perennials” refer to non-fruit-bearing young trees, so their change to fruit-bearing orchard classifications helps us

estimate the age of the orchard. LandIQ crop type data are available for the years 2014, 2016, 2018, 2019, and 2020. Since we have agricultural ET data for the years 2016, 2018, and 2019, we can date orchards that have been bearing fruit for up to five years (i.e. these would be orchards for which we have agricultural ET data in 2019, and were classified as young perennials for the last time in 2014.) Using this technique, we uncover a clear relationship between orchard age and agricultural ET (Supplementary Fig. 7).

Supplementary Fig. 7. Water consumption by life stage of orchards in California’s Central Valley. Prior to bearing fruit, orchards have low agricultural water consumption, or evapotranspiration (ET). Water consumption then increases rapidly during the first 5 years of fruit production, after which it stabilizes.

Given this substantial effect of orchard age on agricultural ET, we enact two changes. First, we include this finding in our initial discussion of within-crop variability in the results section (line 148):

Part of the large variability in agricultural ET for deciduous fruits and nuts can be explained by orchard age: young orchards consume significantly less water than more mature ones (Supplementary Note 2, Supplementary Fig. 7). Climate, topography, and soil quality explain an additional 6% of within-crop variation. However, substantial within-crop variation remains, indicating that some of these differences may be due to variations in farming practices. This would suggest that reducing water consumption without switching crops may be feasible, which we explore in the next section.

Next, we adjust our management scenario simulations to account for orchard age. We describe this change briefly in our results section (line 166):

Because it would not make sense to prescribe farmers in different contexts to consume similar amounts of water, we calculate crop-specific agricultural ET and within-crop variation at the level of groundwater sub-basins. We additionally control for differences in climate, topography, and soil type before conducting the scenarios. To account for the effects of orchard age, we remove orchards that have been bearing fruit for 5 years or less or that are in their last year of production (Supplementary Fig. 7).

A more detailed explanation of the methodology used is described in the methods (line 465):

To ensure that the water savings identified in our scenarios result from factors that can be influenced by management interventions, we account for orchard age, climate, and other physical characteristics of the land. To account for orchard age, we remove young orchards that have been bearing fruit for 5 or less years, or old orchards that are going to be removed within the next year (Supplementary Note 2). Because we are only able to label orchard age in this way for all orchards in year 2019 (the DWR crop type data we use are only available starting 2014), we exclusively use 2019 for this part of the analysis. Since we are only using one year, we do not adjust the water use according to the year like we do for the other analyses. To account for differences in climate and other characteristics inherent to the land, we control for potential ET, soil quality, topographic wetness index, elevation, aspect and slope using linear regression (eq. 5).

While these modifications are essential for reinforcing the validity of our findings, it is noteworthy that the results remain largely consistent, even with the analysis limited to the year 2019.

3.3.2 Limitations of the farming practices scenario

We recognize that even after accounting for differences in orchard age, our analysis cannot account for other equally important factors such as production goals and management style. As a result, we have revisited our discussion of the farmer practices scenario to more clearly recognize these limitations (line 245).

[...] Unfortunately, a lack of high-resolution data on field-scale farming practices and yields inhibits further analysis of specific practices driving agricultural ET variation and their economic implications. However, prior literature from experimental plots or particular locations suggests that mulching (Kader et al., 2019), conservation tillage (Mitchell et al., 2012), deficit irrigation (English and Raja, 1996; Goldhamer, 1999; Rudnick et al., 2019), and improved irrigation scheduling and technologies (Jovanovic et al., 2020) all have potential to limit agricultural ET. These practices may prove advantageous relative to costly strategies like crop-switching or fallowing, though more detailed cost-benefit analyses are necessary to determine the suitability of various interventions in specific contexts (Mitchell et al., 2016). Such water saving farming practices are not mentioned in the plans drafted by SGMA water managers (Bruno et al., 2022). This is possibly due to a lack of conclusive research on the potential of such strategies to effectively decrease agricultural ET without significant effects on yield or operation cost. Spatial data on the use of these different practices would allow researchers to take full advantage of our high-resolution agricultural ET estimates and study their water-saving and economic benefits.

3.4 More specifics

Line 31: This first sentence is a bit confusing – are you saying the overexploitation of ‘drought’ and ‘climate change’ have led to it seems like there is simply some grammatical error tying the sentence together.

Thank you for this point; we have re-organized this sentence to increase clarity (line 32):

Climate change, drought, and the overexploitation of water resources have led to declines in freshwater storage in many vital agricultural regions (Rodell et al., 2018), raising concerns surrounding the future of food and water security (Elliott et al., 2014; Famiglietti, 2014).

Line 43: “because a large share of irrigation water is returned to the system as runoff or recharge”. I understand the statement and it is true, but because the study takes place in California, it is important to note the high efficiency of the irrigation in the state (drip or micro-sprinklers installed). In these systems, very little irrigated water is lost to runoff or deep percolation.

We have edited the statement to include this information (line 42):

While it is possible to monitor volumes of water **withdrawn** for irrigation, this is a poor proxy for the amount of water crops consume through evapotranspiration (ET) (Marston et al., 2022). **Only some irrigation water results in ET: the rest remains in the system as runoff or recharge, though this proportion can vary widely depending on topography, climate, soil type, and farming practices** (Ward and Pulido-Velazquez, 2008).

Line 46: Consider rephrasing in that evapotranspiration is not evaporated by the soil but rather from the soil to the atmosphere.

Thank you for this point; we have rephrased this passage as follows (line 46):

Agricultural ET, or the increase in ET that irrigated agriculture brings, is **therefore** a critical measure in that it represents the amount of water that is actually “consumed” by agriculture. **This water leaves the watershed entirely as it is evaporated from the soil and transpired by crops** (Hoekstra and Mekonnen, 2012).

Line 51: This last sentence is a bit confusing to me. How does decreasing water diversions not affect food production? Because irrigation water does not contribute to crop growth? Maybe I'm missing something?

We apologize for this confusing sentence. What we meant to highlight in this passage is that of all the water used for irrigation, only that which ends up as agricultural ET has the potential to contribute to crop growth, and therefore food production. We have modified this section to improve clarity (line 52):

When compared to total irrigation amounts, agricultural ET can also be used to highlight the fraction of irrigation withdrawals that do not effectively result in consumptive use (Puy et al., 2022). **While this water stays in the system and therefore may continue to provide beneficial uses** (Grafton et al., 2018), **it also does not achieve its original purpose of contributing to crop growth. Therefore, agricultural ET can help calculate irrigation efficiency and identify unnecessary irrigation water withdrawals from surface or groundwater reservoirs** (Puy et al., 2022).

Line 112: Here you give error metrics in cm per year, but you specify they will be mm per year on line 285. I would suggest reporting values in mm per year as I've never seen ET reported in cm per year. It might also be worth mentioning that all growers use acre-in or acre-feet as their unit of choice. I would suggest converting your overall units (mm/day) to acre-in/acre-feet in maybe the conclusion/discussion so a broader readership can more quickly interpret your results.

We apologize for the inconsistency in metrics used, and thank you for your suggestion to use the more common mm per year metric. We have changed all calculations to be represented in mm per year (e.g. line 125, 136, 137). To maximize interpretability of our results

for a broad readership, we do not use specific numbers in the discussion or conclusion and instead focus on the trends and relative values we uncover in our analysis. Therefore, we do not see an opportunity to use acre-in or acre-feet units in the discussion or conclusion, but will be sure to do so if we revise these sections to include specific numbers in the future.

Line 113: Your test set is made up of 2 km² regions – can you mention how many? Also, these regions represent fallowed land. Are all 2 km² regions completely fallowed land? I guess I’m asking if all fallowed pixels chosen are arbitrary 2 km² boundaries placed all over the central valley and not polygons outlining specific fallowed fields.

We apologize for the lack of clarity surrounding our hold-out sets. The 2 km² areas are not polygons outlining specific fallowed fields, but rather arbitrary boundaries. We have clarified our methodology and mention the number of regions in the methods section (line 396):

To validate our naturally-occurring ET model, we split our dataset, reserving 60% for training, 10% for validation, and 30% for testing. In order to ensure that nearby and therefore very similar pixels are not present across multiple splits, we group our splits by 2 km² squares, four times the size of a large agricultural field in the Central Valley. **The entire dataset is made up of over 16 million pixels populating 8,180 distinct 2 km² regions. Nearly 10 million pixels and 4,908 2 km² clusters are contained in the training split, and 2,454 clusters are reserved for testing.**

Line 137: See concern #3. Line 149: See concern #3.

These lines have been revised in accordance with our response to concern #3.

Line 208: Irrelevant and misleading. This 30% figure is from citations looking at wheat, maize, and cotton; all of which are annuals and all of which have no notable economic viability in California (combined, their economic output is only 2% of almonds and grapes combined).

Thank you for flagging this. Given that this statement is not relevant to the California context, we have removed this statement and replaced it with a more appropriate discussion about different farmer practices as described in our response to your third comment (line 236).

Line 250: Sorry for likely not following, but isn’t this 7% much lower than values specified earlier? Where is this coming from?

We apologize for the confusion surrounding this statement, “Finally, the naturally-occurring ET estimates have a MAE of 3.6 cm per year, representing only 7% of the average agricultural ET of an active agricultural field.” The 7% refers to the mean absolute error of our machine learning model that predicts naturally-occurring ET, not naturally-occurring

ET itself. We have removed this statement to stave off confusion and have replaced it with a more straightforward description of error (line 308):

Additionally, the OpenET ensemble model has been extensively validated (Melton et al., 2022; OpenET, 2021), and our machine learning model has an R^2 of .87. We estimate that these sources of error are responsible for only 11% of the variance in our yearly agricultural ET estimates (Supplementary Note 4). Nevertheless, the water-saving potentials we calculate for both the fallowing and farmers practice scenarios should be interpreted as upper bound estimates.

References

- Aires, L., Pio, C., and Pereira, J. (2008). The effect of drought on energy and water vapour exchange above a mediterranean c3/c4 grassland in southern portugal. *Agricultural and Forest Meteorology*, 148(4):565–579.
- Al-Gaadi, K. A., Madugundu, R., Tola, E., El-Hendawy, S., and Marey, S. (2022). Satellite-Based Determination of the Water Footprint of Carrots and Onions Grown in the Arid Climate of Saudi Arabia. *Remote Sensing*, 14(23):5962. Number: 23 Publisher: Multidisciplinary Digital Publishing Institute.
- Anderson, M., Gao, F., Knipper, K., Hain, C., Dulaney, W., Baldocchi, D., Eichelmann, E., Hemes, K., Yang, Y., Medellin-Azuara, J., and Kustas, W. (2018). Field-Scale Assessment of Land and Water Use Change over the California Delta Using Remote Sensing. *Remote Sensing*, 10(6):889. Number: 6 Publisher: Multidisciplinary Digital Publishing Institute.
- Anderson, M., Yang, Y., Xue, J., Knipper, K. R., Yang, Y., Gao, F., Hain, C. R., Kustas, W. P., Cawse-Nicholson, K., Hulley, G., Fisher, J. B., Alfieri, J. G., Meyers, T. P., Prueger, J., Baldocchi, D. D., and Rey-Sanchez, C. (2021). Interoperability of ECOSTRESS and Landsat for mapping evapotranspiration time series at sub-field scales. *Remote Sensing of Environment*, 252:112189.
- Baldocchi, D. D. (2003). Assessing the eddy covariance technique for evaluating carbon dioxide exchange rates of ecosystems: past, present and future. *Global Change Biology*, 9(4):479–492. eprint: <https://onlinelibrary.wiley.com/doi/pdf/10.1046/j.1365-2486.2003.00629.x>.
- Baldocchi, D. D., Xu, L., and Kiang, N. (2004). How plant functional-type, weather, seasonal drought, and soil physical properties alter water and energy fluxes of an oak–grass savanna and an annual grassland. *Agricultural and Forest Meteorology*, 123(1):13–39.
- Bruno, E. M., Hagerty, N., and Wardle, A. R. (2022). The political economy of groundwater management: Descriptive evidence from california. In Libecap, G. D. and Dinar, A., editors, *American Agriculture, Water Resources, and Climate Change*, chapter Chapter Number. University of Chicago Press.
- Drechsler, K., Fulton, A., and Kisekka, I. (2022). Crop coefficients and water use of young almond orchards. *Irrigation Science*, 40(3):379–395.
- Duell, L. F. W. (1988). Estimates of evapotranspiration in alkaline scrub and meadow communities of owens valley, california, using the bowen-ratio, eddy-correlation, and penman-combination methods.
- Elliott, J., Deryng, D., Müller, C., Frieler, K., Konzmann, M., Gerten, D., Glotter, M., Flörke, M., Wada, Y., Best, N., Eisner, S., Fekete, B. M., Folberth, C., Foster, I., Gosling, S. N., Haddeland, I., Khabarov, N., Ludwig, F., Masaki, Y., Olin, S., Rosenzweig, C.,

- Ruane, A. C., Satoh, Y., Schmid, E., Stacke, T., Tang, Q., and Wisser, D. (2014). Constraints and potentials of future irrigation water availability on agricultural production under climate change. *Proceedings of the National Academy of Sciences*, 111(9):3239–3244. Publisher: Proceedings of the National Academy of Sciences.
- English, M. and Raja, S. N. (1996). Perspectives on deficit irrigation. *Agricultural Water Management*, 32(1):1–14.
- Famiglietti, J. S. (2014). The global groundwater crisis. *Nature Climate Change*, 4(11):945–948. Number: 11 Publisher: Nature Publishing Group.
- Fernandez-Bou, A. S., Rodríguez-Flores, J. M., Guzman, A., Ortiz-Partida, J. P., Classen-Rodriguez, L. M., Sánchez-Pérez, P. A., Valero-Fandiño, J., Pells, C., Flores-Landeros, H., Sandoval-Solís, S., Characklis, G. W., Harmon, T. C., McCullough, M., and Medellín-Azuara, J. (2023). Water, environment, and socioeconomic justice in California: A multi-benefit cropland repurposing framework. *Science of The Total Environment*, 858:159963.
- Goldhamer, D. A. (1999). Regulated deficit irrigation for California canning olives. *Acta Horticulturae*, (474):369–372.
- Grafton, R. Q., Williams, J., Perry, C. J., Molle, F., Ringler, C., Steduto, P., Udall, B., Wheeler, S. A., Wang, Y., Garrick, D., and Allen, R. G. (2018). The paradox of irrigation efficiency. *Science*, 361(6404):748–750. Publisher: American Association for the Advancement of Science.
- Guivetchi, K., Goyal, A., Wiekling, J., Marr, J., Dhillon, D., Arrate, D., Stygar, S., and Maendly, R. (2018). FLOOD-MAR: Using Flood Water for Managed Aquifer Recharge to Support Sustainable Water Resources.
- Hanak, E., Lund, J., Arnold, B., Escriva-Bou, A., Gray, B., Green, S., Harter, T., Howitt, R., MacEwan, D., Medellín-Azuara, J., Moyle, P., and Seavy, N. (2017). Water Stress and a Changing San Joaquin Valley. *Public Policy Institute of California*, page 50.
- Hoekstra, A. Y. and Mekonnen, M. M. (2012). The water footprint of humanity. *Proceedings of the National Academy of Sciences*, 109(9):3232–3237. Publisher: Proceedings of the National Academy of Sciences.
- Hornbeck, R. (2012). The Enduring Impact of the American Dust Bowl: Short- and Long-Run Adjustments to Environmental Catastrophe. *American Economic Review*, 102(4):1477–1507.
- Johansson, E. L., Fader, M., Seaquist, J. W., and Nicholas, K. A. (2016). Green and blue water demand from large-scale land acquisitions in Africa. *Proceedings of the National Academy of Sciences*, 113(41):11471–11476. Publisher: Proceedings of the National Academy of Sciences.
- Jovanovic, N., Pereira, L. S., Paredes, P., Pôças, I., Cantore, V., and Todorovic, M. (2020).

- A review of strategies, methods and technologies to reduce non-beneficial consumptive water use on farms considering the FAO56 methods. *Agricultural Water Management*, 239:106267.
- Kader, M. A., Singha, A., Begum, M. A., Jewel, A., Khan, F. H., and Khan, N. I. (2019). Mulching as water-saving technique in dryland agriculture: review article. *Bulletin of the National Research Centre*, 43(1):147.
- Kelsey, R., Hart, A., Butterfield, H. S., and Dink, V. (2018). Groundwater sustainability in the San Joaquin Valley: Multiple benefits if agricultural lands are retired and restored strategically. *California Agriculture*, 72(3).
- Kurukulasuriya, P. and Mendelsohn, R. (2008). Crop switching as a strategy for adapting to climate change. *African Journal of Agricultural and Resource Economics*. Num Pages: 22.
- LandIQ (2021). New Land Use Resource Publicly Available.
- Mancosu, N., Spano, D., Orang, M., Sarreshteh, S., and Snyder, R. L. (2016). SIMETAW# - a Model for Agricultural Water Demand Planning. *Water Resources Management*, 30(2):541–557.
- Marston, L. T., Abdallah, A. M., Bagstad, K. J., Dickson, K., Glynn, P., Larsen, S. G., Melton, F. S., Onda, K., Painter, J. A., Prairie, J., Ruddell, B. L., Rushforth, R. R., Senay, G. B., and Shaffer, K. (2022). Water-Use Data in the United States: Challenges and Future Directions. *JAWRA Journal of the American Water Resources Association*, 58(4):485–495. _eprint: <https://onlinelibrary.wiley.com/doi/pdf/10.1111/1752-1688.13004>.
- Melton, F. S., Huntington, J., Grimm, R., Herring, J., Hall, M., Rollison, D., Erickson, T., Allen, R., Anderson, M., Fisher, J. B., Kilic, A., Senay, G. B., Volk, J., Hain, C., Johnson, L., Ruhoff, A., Blankenau, P., Bromley, M., Carrara, W., Daudert, B., Doherty, C., Dunkerly, C., Friedrichs, M., Guzman, A., Halverson, G., Hansen, J., Harding, J., Kang, Y., Ketchum, D., Minor, B., Morton, C., Ortega-Salazar, S., Ott, T., Ozdogan, M., ReVelle, P. M., Schull, M., Wang, C., Yang, Y., and Anderson, R. G. (2022). OpenET: Filling a Critical Data Gap in Water Management for the Western United States. *JAWRA Journal of the American Water Resources Association*, 58(6):971–994. _eprint: <https://onlinelibrary.wiley.com/doi/pdf/10.1111/1752-1688.12956>.
- Mitchell, J., Singh, P., Wallender, W., Munk, D., Wroble, J., Horwath, W., Hogan, P., Roy, R., and Hanson, B. (2012). No-tillage and high-residue practices reduce soil water evaporation. *California Agriculture*, 66(2):55–61. Publisher: University of California, Agriculture and Natural Resources.
- Mitchell, J. P., Harben, R., Sposito, G., Shrestha, A., Munk, D. S., Miyao, G., Southard, R., Ferris, H., Horwath, W. R., Kueneman, E., Fisher, J., Bottens, M., Hogan, P., Roy, R., Komar, J., Beck, D., Reicosky, D., Leinfelder-Miles, M., Aegerter, B. J., Six, J., Barcellos,

- T., Giacomazzi, D., Sano, A., Sanchez, J., Crowell, M., Diener, J., Cordova, D., Cordova, T., and Rossiter, J. (2016). Conservation agriculture: Systems thinking for sustainable farming. *California Agriculture*, 70(2).
- National Academies of Sciences, Engineering, and Medicine. (2019). *Thriving on Our Changing Planet: A Decadal Strategy for Earth Observation from Space*. National Academies Press. Google-Books-ID: dumADwAAQBAJ.
- Nelson, K. S. and Burchfield, E. K. (2017). Effects of the Structure of Water Rights on Agricultural Production During Drought: A Spatiotemporal Analysis of California's Central Valley. *Water Resources Research*, 53(10):8293–8309. eprint: <https://onlinelibrary.wiley.com/doi/pdf/10.1002/2017WR020666>.
- OpenET (2021). Intercomparison and Accuracy Report.
- Orang, M. N., Snyder, R. L., Shu, G., Hart, Q. J., Sarreshteh, S., Falk, M., Beaudette, D., Hayes, S., and Eching, S. (2013). California Simulation of Evapotranspiration of Applied Water and Agricultural Energy Use in California. *Journal of Integrative Agriculture*, 12(8):1371–1388.
- Pastorello, G., Trotta, C., Canfora, E., Chu, H., Christianson, D., Cheah, Y.-W., Poindexter, C., Chen, J., Elbashandy, A., Humphrey, M., Isaac, P., Polidori, D., Reichstein, M., Ribeca, A., van Ingen, C., Vuichard, N., Zhang, L., Amiro, B., Ammann, C., Arain, M. A., Ardö, J., Arkebauer, T., Arndt, S. K., Arriga, N., Aubinet, M., Aurela, M., Baldocchi, D., Barr, A., Beamesderfer, E., Marchesini, L. B., Bergeron, O., Beringer, J., Bernhofer, C., Berveiller, D., Billesbach, D., Black, T. A., Blanken, P. D., Bohrer, G., Boike, J., Bolstad, P. V., Bonal, D., Bonnefond, J.-M., Bowling, D. R., Bracho, R., Brodeur, J., Brümmer, C., Buchmann, N., Burban, B., Burns, S. P., Buysse, P., Cale, P., Cavagna, M., Cellier, P., Chen, S., Chini, I., Christensen, T. R., Cleverly, J., Collalti, A., Consalvo, C., Cook, B. D., Cook, D., Coursolle, C., Cremonese, E., Curtis, P. S., D'Andrea, E., da Rocha, H., Dai, X., Davis, K. J., Cinti, B. D., Grandcourt, A. d., Ligne, A. D., De Oliveira, R. C., Delpierre, N., Desai, A. R., Di Bella, C. M., Tommasi, P. d., Dolman, H., Domingo, F., Dong, G., Dore, S., Duce, P., Dufrêne, E., Dunn, A., Dušek, J., Eamus, D., Eichelmann, U., ElKhidir, H. A. M., Eugster, W., Ewenz, C. M., Ewers, B., Famulari, D., Fares, S., Feigenwinter, I., Feitz, A., Fensholt, R., Filippa, G., Fischer, M., Frank, J., Galvagno, M., Gharun, M., Gianelle, D., Gielen, B., Gioli, B., Gitelson, A., Goded, I., Goekede, M., Goldstein, A. H., Gough, C. M., Goulden, M. L., Graf, A., Griebel, A., Gruening, C., Grünwald, T., Hammerle, A., Han, S., Han, X., Hansen, B. U., Hanson, C., Hatakka, J., He, Y., Hehn, M., Heinesch, B., Hinko-Najera, N., Hörtnagl, L., Hutley, L., Ibrom, A., Ikawa, H., Jackowicz-Korczynski, M., Janouš, D., Jans, W., Jassal, R., Jiang, S., Kato, T., Khomik, M., Klatt, J., Knohl, A., Knox, S., Kobayashi, H., Koerber, G., Kolle, O., Kosugi, Y., Kotani, A., Kowalski, A., Kruijt, B., Kurbatova, J., Kutsch, W. L., Kwon, H., Launiainen, S., Laurila, T., Law, B., Leuning, R., Li, Y., Liddell, M., Limousin, J.-M., Lion, M., Liska, A. J., Lohila, A., López-Ballesteros, A., López-Blanco, E., Loubet, B., Loustau, D., Lucas-Moffat, A., Lüers, J., Ma, S., Macfarlane, C., Magliulo, V., Maier,

- R., Mammarella, I., Manca, G., Marcolla, B., Margolis, H. A., Marras, S., Massman, W., Mastepanov, M., Matamala, R., Matthes, J. H., Mazzenga, F., McCaughey, H., McHugh, I., McMillan, A. M. S., Merbold, L., Meyer, W., Meyers, T., Miller, S. D., Minerbi, S., Moderow, U., Monson, R. K., Montagnani, L., Moore, C. E., Moors, E., Moreaux, V., Moureaux, C., Munger, J. W., Nakai, T., Neiryneck, J., Nesic, Z., Nicolini, G., Noormets, A., Northwood, M., Nosetto, M., Nouvellon, Y., Novick, K., Oechel, W., Olesen, J. E., Ourcival, J.-M., Papuga, S. A., Parmentier, F.-J., Paul-Limoges, E., Pavelka, M., Peichl, M., Pendall, E., Phillips, R. P., Pilegaard, K., Pirk, N., Posse, G., Powell, T., Prasse, H., Prober, S. M., Rambal, S., Rannik, , Raz-Yaseef, N., Rebmann, C., Reed, D., Dios, V. R. d., Restrepo-Coupe, N., Reverter, B. R., Roland, M., Sabbatini, S., Sachs, T., Saleska, S. R., Sánchez-Cañete, E. P., Sanchez-Mejia, Z. M., Schmid, H. P., Schmidt, M., Schneider, K., Schrader, F., Schroder, I., Scott, R. L., Sedlák, P., Serrano-Ortíz, P., Shao, C., Shi, P., Shironya, I., Siebicke, L., Šigut, L., Silberstein, R., Sirca, C., Spano, D., Steinbrecher, R., Stevens, R. M., Sturtevant, C., Suyker, A., Tagesson, T., Takanashi, S., Tang, Y., Tapper, N., Thom, J., Tomassucci, M., Tuovinen, J.-P., Urbanski, S., Valentini, R., van der Molen, M., van Gorsel, E., van Huissteden, K., Varlagin, A., Verfaillie, J., Vesala, T., Vincke, C., Vitale, D., Vygodskaya, N., Walker, J. P., Walter-Shea, E., Wang, H., Weber, R., Westermann, S., Wille, C., Wofsy, S., Wohlfahrt, G., Wolf, S., Woodgate, W., Li, Y., Zampedri, R., Zhang, J., Zhou, G., Zona, D., Agarwal, D., Biraud, S., Torn, M., and Papale, D. (2020). The FLUXNET2015 dataset and the ONEFlux processing pipeline for eddy covariance data. *Scientific Data*, 7(1):225. Number: 1 Publisher: Nature Publishing Group.
- Peterson, C., Pittelkow, C., and Lundy, M. (2022). Exploring the Potential for Water-Limited Agriculture in the San Joaquin Valley. *Public Policy Institute of California*.
- Puy, A., Sheikholeslami, R., Gupta, H. V., Hall, J. W., Lankford, B., Lo Piano, S., Meier, J., Pappenberger, F., Porporato, A., Vico, G., and Saltelli, A. (2022). The delusive accuracy of global irrigation water withdrawal estimates. *Nature Communications*, 13(1):3183. Number: 1 Publisher: Nature Publishing Group.
- Rodell, M., Famiglietti, J. S., Wiese, D. N., Reager, J. T., Beaudoin, H. K., Landerer, F. W., and Lo, M.-H. (2018). Emerging trends in global freshwater availability. *Nature*, 557(7707):651–659. Number: 7707 Publisher: Nature Publishing Group.
- Rudnick, D., Irmak, S., West, C., Chávez, J., Kisekka, I., Marek, T., Schneekloth, J., Mitchell McCallister, D., Sharma, V., Djaman, K., Aguilar, J., Schipanski, M., Rogers, D., and Schlegel, A. (2019). Deficit Irrigation Management of Maize in the High Plains Aquifer Region: A Review. *JAWRA Journal of the American Water Resources Association*, 55(1):38–55. eprint: <https://onlinelibrary.wiley.com/doi/pdf/10.1111/1752-1688.12723>.
- Schauer, M. and Senay, G. B. (2019). Characterizing Crop Water Use Dynamics in the Central Valley of California Using Landsat-Derived Evapotranspiration. *Remote Sensing*, 11(15):1782. Number: 15 Publisher: Multidisciplinary Digital Publishing Institute.
- Singer, M. B., Asfaw, D. T., Rosolem, R., Cuthbert, M. O., Miralles, D. G., MacLeod, D.,

- Quichimbo, E. A., and Michaelides, K. (2021). Hourly potential evapotranspiration at 0.1° resolution for the global land surface from 1981-present. *Scientific Data*, 8(1):224. Bandiera_abtest: a Cc_license_type: cc_publicdomain Cg_type: Nature Research Journals Number: 1 Primary_atype: Research Publisher: Nature Publishing Group Subject_term: Hydrology Subject_term_id: hydrology.
- Snyder, R. L., Geng, S., Orang, M., and Sarreshteh, S. (2012). Calculation and Simulation of Evapotranspiration of Applied Water. *Journal of Integrative Agriculture*, 11(3):489–501.
- Ward, F. A. and Pulido-Velazquez, M. (2008). Water conservation in irrigation can increase water use. *Proceedings of the National Academy of Sciences*, 105(47):18215–18220. Publisher: Proceedings of the National Academy of Sciences.
- Wong, A. J., Jin, Y., Medellín-Azuara, J., Paw U, K. T., Kent, E. R., Clay, J. M., Gao, F., Fisher, J. B., Rivera, G., Lee, C. M., Hemes, K. S., Eichelmann, E., Baldocchi, D. D., and Hook, S. J. (2021). Multiscale Assessment of Agricultural Consumptive Water Use in California’s Central Valley. *Water Resources Research*, 57(9):e2020WR028876. eprint: <https://onlinelibrary.wiley.com/doi/pdf/10.1029/2020WR028876>.

REVIEWERS' COMMENTS

Reviewer #1 (Remarks to the Author):

The authors have addressed my questions/feedback satisfactorily. I do not have further requests for revision.

Reviewer #3 (Remarks to the Author):

Many thanks to the author(s) for their revisions. This was a very strong paper prior to revisions and I believe it is even stronger now. The revisions were probably the most thought out responses I have ever received. This is what the scientific (and review) process is supposed to look like and I want to thank the author(s) for their efforts and restoring my confidence in the review process. It was a joy to read the manuscript and I am looking forward to its publication and being able to cite the work.